# Learning "What-if" Explanations for Sequential Decision-Making

**Ioana Bica**
University of Oxford, Oxford, UK
The Alan Turing Institute, London, UK
ioana.bica@eng.ox.ac.uk

**Daniel Jarrett**
University of Cambridge, Cambridge, UK
daniel.jarrett@maths.cam.ac.uk

**Alihan Hüyük**
University of Cambridge, Cambridge, UK
ah2075@cam.ac.uk

**Mihaela van der Schaar**
University of Cambridge, Cambridge, UK
Cambridge Center for AI in Medicine, UK
University of California, Los Angeles, USA
The Alan Turing Institute, London, UK
mv472@cam.ac.uk

## Abstract

Building interpretable parameterizations of real-world decision-making on the basis of demonstrated behavior—i.e. trajectories of observations and actions made by an expert maximizing some unknown reward function—is essential for introspecting and auditing policies in different institutions. In this paper, we propose learning explanations of expert decisions by modeling their reward function in terms of preferences with respect to "what if" outcomes: Given the current history of observations, *what would happen if we took a particular action*? To learn these cost-benefit tradeoffs associated with the expert's actions, we integrate counterfactual reasoning into batch inverse reinforcement learning. This offers a principled way of defining reward functions and explaining expert behavior, and also satisfies the constraints of real-world decision-making—where active experimentation is often impossible (e.g. in healthcare). Additionally, by estimating the effects of different actions, counterfactuals readily tackle the *off-policy* nature of policy evaluation in the batch setting, and can naturally accommodate settings where the expert policies depend on *histories* of observations rather than just current states. Through illustrative experiments in both real and simulated medical environments, we highlight the effectiveness of our batch, counterfactual inverse reinforcement learning approach in recovering accurate and interpretable descriptions of behavior.

## 1 Introduction

Consider the problem of explaining sequential decision-making on the basis of demonstrated behavior. In healthcare, an important goal lies in being able to obtain an interpretable parameterization of the experts' behavior (e.g in terms of how they assign treatments) such that we can quantify and inspect policies in different institutions and uncover the trade-offs and preferences associated with expert actions (James & Hammond, 2000; Westert et al., 2018; Van Parys & Skinner, 2016; Jarrett & van der Schaar, 2020). Moreover, modeling the reward function of different clinical practitioners can be revealing as to their tendencies to treat various diseases more/less aggressively (Rysavy et al., 2015), which —in combination with patient outcomes—has the potential to inform and update clinical guidelines.

In many settings, such as medicine, decision-makers can be modeled as reasoning about "what-if" patient outcomes: Given the available information about the patient, what would happen if we took a particular action? (Djulbegovic et al., 2018; McGrath, 2009). As treatments often affect several patient covariates, by having both benefits and side-effects, decision-makers often make choices based on their preferences over these counterfactual outcomes. Thus, in our case, an interpretable explanation of a policy is one where the reward signal for (sequential) actions is parameterized on the basis of preferences over (sequential) counterfactuals (i.e. "what-if" patient outcomes).

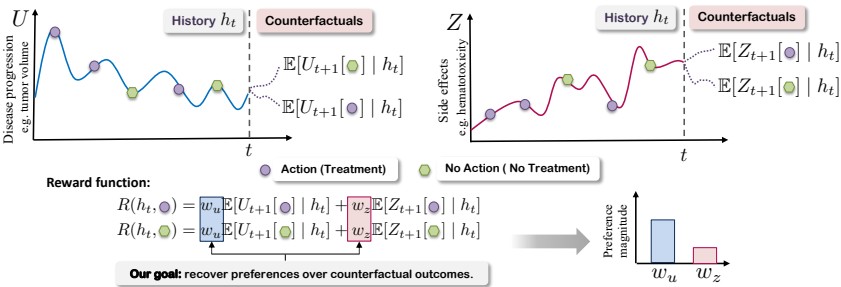

Figure 1: Explaining decision-making behaviour in terms of preferences over "what if" outcomes. Consider the evolution of tumour volume ($U$) and side effects ($Z$) under a binary action. $\mathbb{E}[U_{t+1}[a_t] \mid h_t]$ and $\mathbb{E}[Z_{t+1}[a_t] \mid h_t]$ are the counterfactuals for the patient features under action $a_t$ given history $h_t$ of prior actions and covariates. Parameterizing the reward as the weighted sum of these counterfactuals: $R(h_t, a_t) = w_u \mathbb{E}[U_{t+1}[a_t] \mid h_t] + w_z \mathbb{E}[Z_{t+1}[a_t] \mid h_t]$, naturally allows us to model the preferences of experts: e.g. finding that $|w_u| > |w_z|$ indicates that the expert is treating more aggressively, by placing more weight on reducing tumour volume than on minimizing side effects.

Given the observations and actions made by an expert, *inverse reinforcement learning* (IRL) offers a principled way for modeling their behavior by recovering the (unknown) reward function being maximized (Ng et al., 2000; Abbeel & Ng, 2004; Choi & Kim, 2011). Standard solutions operate by iterating on candidate reward functions, solving the associated (forward) reinforcement learning problem at each step. In many real-world problems, however, we are specifically interested in the challenge of offline learning—that is, where further experimentation is not possible—such as in medicine. In this *batch* setting, we only have access to trajectories sampled from the expert policy in the form of an observational dataset—such as in electronic health records.

**Batch IRL.** By their nature, classic IRL algorithms require interactive access to the environment, or full knowledge of the environment's dynamics (Ng et al., 2000; Abbeel & Ng, 2004; Choi & Kim, 2011). While batch IRL solutions have been proposed by way of off-policy evaluation (Klein et al., 2011; 2012; Lee et al., 2019), they suffer from two disadvantages. First, they are limited by the assumption that state dynamics are fully-observable and Markovian. This is hardly true in medicine: treatment assignments generally depend on how patient covariates have evolved over time (Futoma et al., 2020). Second, rewards are often parameterized as uninterpretable representations of neural network hidden states and consequently cannot be used to explain sequential decision making.

**"What-if" Explanations.** To address these shortcomings and to obtain a parameterizable interpretation of the expert's behavior, we propose explicitly incorporating counterfactual reasoning into batch IRL. In particular, we focus on "what if" explanations for modeling decision-making, while simultaneously accounting for the partially-observable nature of patient histories. Under the max-margin apprenticeship framework (Abbeel & Ng, 2004; Klein et al., 2011; Lee et al., 2019), we learn a parameterized reward function $R(h_t, a_t)$ that is defined as a weighted sum over *potential outcomes* (Rubin, 2005) for taking action $a_t$ given history $h_t$.

As highlighted in Figure 1, consider the decision making process of assigning a binary action given the tumour volume ($U$) and side effects ($Z$). Let $\mathbb{E}[U_{t+1}[a_t] \mid h_t]$ and $\mathbb{E}[Z_{t+1}[a_t] \mid h_t]$ be the counterfactual outcomes for the two covariates when action $a_t$ is taken given the history $h_t$ of covariates and previous actions. We define the reward as the weighted sum of these counterfactuals: $R(h_t, a_t) = w_u \mathbb{E}[U_{t+1}[a_t] \mid h_t] + w_z \mathbb{E}[Z_{t+1}[a_t] \mid h_t]$, to take into account the effect of actions and to directly model the preferences of the expert. The ideal scenario is when both the tumour volume and the side effects are zero, so the reward weights of a doctor aiming for this are both negative. However, recovering that $|w_u| > |w_z|$, it means that the doctor is treating more aggressively, as they are focusing more on reducing the tumour volume rather than on the side effects of treatments. Alternatively, $|w_u| < |w_z|$ indicates that the side effects are more important and the expert is treating less aggressively. Our motivation for using counterfactuals to define the reward comes from the idea that rational decision making considers the potential effects of actions (Djulbegovic et al., 2018).

**Contributions.** Exploring the synergy between counterfactual reasoning and batch IRL for understanding sequential decision making confers multiple advantages. First, it offers a principled approach for parameterizing reward functions in terms of preferences over *what-if* patient outcomes, which enables us to explain the cost-benefit tradeoffs associated with an expert's actions. Second, by estimating the effects of different actions, counterfactuals readily tackle the *off-policy* nature of

| Method | Environment | Batch | Feature map for reward | Policy | Feat. expectations |
|---|---|---|---|---|---|
| Abbeel & Ng (2004) | Model-based | No | $\phi(s_t)$ = basis functions for state $s_t$ | $\pi(a_t \mid s_t)$ | Model roll-outs |
| Choi & Kim (2011) | Model-based | No | $\sum_s b_t(s)\phi(s, a_t)$ = basis for belief $b_t$ | $\pi(a_t \mid b_t)$ | Model roll-outs |
| Klein et al. (2011) | Model-free | Yes | $\phi(x_t)$ = basis functions for state $x_t$ | $\pi(a_t \mid x_t)$ | LSTD-Q |
| Lee et al. (2019) | Model-free | Yes | $\phi(x_t, a_t) = \text{concat}(\phi(x_t), a_t)$ | $\pi(a_t \mid x_t)$ | DSFN |
| Ours | Model-free | Yes | $\phi(h_t, a_t) = \mathbb{E}[Y_{t+1}[a_t]|h_t]$ | $\pi(a_t \mid h_t)$ | Counterfactual $\mu$-learning |

Table 1: Comparison of our proposed method (batch, counterfactual IRL) with related works in IRL.

policy evaluation in the batch setting. Furthermore, we demonstrate that not only does this alleviate the *cold-start* problem typical of conventional batch IRL solutions, but also accommodates settings where the usual assumption of full observability fails to hold. Through experiments in both real and simulated medical environments, we illustrate the effectiveness of our batch, counterfactual inverse reinforcement learning approach in recovering accurate and interpretable descriptions of behavior.

## 2 RELATED WORKS

In our work, the aim is to explain decision-making by recovering the preferences of experts with respect to the effects of their actions, denoted by the counterfactual outcomes. This goal is fundamentally different from the goal of IRL methods which generally aim to match the performance of experts. We operate under the standard max-margin apprenticeship framework (Ng et al., 2000; Abbeel & Ng, 2004), which searches for a reward function that minimizes the margin between feature expectations of the expert and candidate policies. However, our approach to recovering and understanding decision policies is uniquely characterized by incorporating counterfactuals to obtain explainable reward functions. To tackle the challenges posed by real-world decision making, our method also operates in an offline and model-free manner, and accommodates partially-observable environments.

*Explainability*. By using basis functions (Klein et al., 2012) or hidden layers of a deep network (Lee et al., 2019) to define the feature map, the learned rewards of either approach are inherently uninterpretable, and cannot be used to explain differences in expert behavior. An alternative approach for recovering the expert policy (without reward functions) is imitation learning (Hussein et al., 2017; Osa et al., 2018; Torabi et al., 2019; Jarrett et al., 2020). However, these methods do not allow us to fully model the decision-making process of experts and to uncover the trade-offs behind their actions.

*Batch Learning*. Klein et al. (2011) propose an off-policy evaluation method based on least squares temporal difference (LSTD-$Q$) (Lagoudakis & Parr, 2003) for estimating feature expectations, and Klein et al. (2012) use a linear score-based classifier to directly approximate the $Q$-function offline. However, both methods require the constraining assumptions that rewards are direct, linear functions of fully-observable states—assumptions we cannot afford to make in realistic settings such as medicine. Lee et al. (2019) propose a deep successor feature network (DSFN) based on $Q$-learning to estimate feature expectations. But their approach similarly assumes fully-observable states, and additionally suffers from the "cold-start" problem where off-policy evaluations are heavily biased unless the initial candidate policy is (already) close to the expert.

*Partial Observability*. No existing batch IRL method accommodates modeling expert policies that depend on patient histories. While Choi & Kim (2011) and Makino & Takeuchi (2012) extend the apprenticeship learning paradigm to partially observable environments by considering policies on beliefs over states, both need to interact with the environment (or a perfect simulator) during learning.

To the best of our knowledge, we are the first to propose explaining sequential decisions through counterfactual reasoning and to tackle the batch IRL problem in partially-observable environments. Our use of the estimated counterfactuals yields inherently interpretable rewards and simultaneously addresses the cold-start problem in Lee et al. (2019). Table 1 highlights the main differences between our method and the relevant related works. See Appendix A for additional related works.

## 3 PROBLEM FORMULATION

**Preliminaries.** At timestep $t$, let random variable $X_t \in \mathcal{X}$ denote the observed patient features and let $A_t \in \mathcal{A}$ denote the action (e.g. treatment) taken, where $\mathcal{A}$ is a finite set of actions. Let $x_t$ and $a_t$ denote realizations of these random variables. Let $h_t = (x_0, a_0, \ldots, x_{t-1}, a_{t-1}, x_t) = (x_{0:t}, a_{0:t-1}) \in \mathcal{H}$ be a realization of the history $H_t \in \mathcal{H}$ of patient observations and actions until timestep $t$.

A stationary stochastic policy represents a mapping: $\pi : \mathcal{H} \times \mathcal{A} \to [0, 1]$, where $\pi(a \mid h)$ indicates the probability of choosing action $a \in \mathcal{A}$ given history $h \in \mathcal{H}$ and $\sum_{a \in \mathcal{A}} \pi(a \mid h) = 1$. Taking action $a_t$ under history $h_t$ results in observing $x_{t+1}$ and obtaining $h_{t+1}$. The reward function is $R : \mathcal{H} \times \mathcal{A} \to \mathbb{R}$ where $R(h, a)$ represents the reward for taking action $a \in \mathcal{A}$ given history $h \in \mathcal{H}$. The value function of a policy $\pi$, $V : \mathcal{H} \to \mathbb{R}$ is defined as: $V^\pi(h) = \mathbb{E}[\sum_{t=0}^\infty \gamma^t R(H_t, A_t) \mid \pi, H_0 = h]$, where $\gamma \in [0, 1)$ is the discount factor and $A_t \sim \pi(\cdot \mid H_t)$ for $t \geq 0$. The action-value function $Q : \mathcal{H} \times \mathcal{A} \to \mathbb{R}$ of a policy is defined as $Q^\pi(h, a) = \mathbb{E}[\sum_{t=0}^\infty \gamma^t R(H_t, A_t) \mid \pi, H_0 = h, A_0 = a]$ where $A_t \sim \pi(\cdot \mid H_t)$ for $t \geq 0$. A higher $Q$-value indicates that action $a$ will yield better long term returns if taken for history $h$. We assume we know the discount factor $\gamma$ which indicates the importance of future rewards for the current history and action pair.

**Batch IRL.** Let $\mathcal{D} = \{\zeta^i\}_{i=1}^N$ be a batch observational dataset consisting of $N$ patient trajectories: $\zeta^i = (x_0^i, a_0^i, \dots x_{T^i-1}^i, a_{T^i-1}^i, x_{T^i}^i)$. The trajectory $\zeta^i$ for patient $i$ consists of covariates $x_t^i$ and actions $a_t^i$ observed for $T^i$ timesteps. For simplicity, we drop the superscript $i$ unless explicitly needed. The actions $a_t \in \mathcal{D}$ are assigned according to some expert policy $\pi_E$ such that $a_t \sim \pi_E(\cdot \mid h_t)$.

We work in the apprenticeship learning set-up (Abbeel & Ng, 2004) and we consider a linear reward function $R(h_t, a_t) = w \cdot \phi(h_t, a_t)$, where the weights $w \in \mathbb{R}^d$ satisfy $\|w\|_1 \leq 1$. The feature map $\phi : \mathcal{H} \times A \to \mathbb{R}^d$ also satisfies $\|\phi(\cdot)\|_2 \leq 1$ such that the reward is bounded. We assume that the expert policy $\pi_E$ is attempting to optimize, without necessarily succeeding, some unknown reward function $R^*(h_t, a_t) = w^* \cdot \phi(h_t, a_t)$, where $w^*$ are the 'true' reward weights. Given $R(h_t, a_t)$, the value of policy $\pi$ can be re-written as: $\mathbb{E}[V^\pi(H_0)] = \mathbb{E}[\sum_{t=0}^\infty \gamma^t w \cdot \phi(H_t, A_t) \mid \pi] = w \cdot \mathbb{E}[\sum_{t=0}^\infty \gamma^t \phi(H_t, A_t) \mid \pi]$, where the expectation is taken with respect to the sequence of histories and action pairs $(H_t, A_t)_{t \geq 0}$ obtained by acting according to $\pi$. The feature expectation of policy $\pi$, defined as the expected discounted cumulative feature vector obtained when choosing actions according to policy $\pi$ is $\mu^\pi = \mathbb{E}[\sum_{t=0}^\infty \gamma^t \phi(H_t, A_t) \mid \pi] \in \mathbb{R}^d$ such that: $\mathbb{E}[V^\pi(H_0)] = w \cdot \mu^\pi$.

Our aim is to recover the expert weights $w^*$ as well as find a policy $\pi$ that is close to the policy of the expert $\pi_E$. We take the max-margin IRL approach and we measure the similarity between the feature expectations of the expert's policy and the feature expectations of a candidate policy using $\|\mu^{\pi_E} - \mu^\pi\|_2$. In this batch IRL setting, we do not have knowledge of transition dynamics and we cannot sample more trajectories from the environment. Note that in this context, we are the first to model expert policies that depend on patient histories and not just current observations.

**Counterfactual reasoning.** To explain the expert's behaviour in terms of their trade-off associated with "what if" outcomes, we use counterfactual reasoning to define the feature map $\phi(h_t, a_t)$ part of the reward $R(h_t, a_t) = w \cdot \phi(h_t, a_t)$. We adopt the potential outcomes framework (Neyman, 1923; Rubin, 1978; Robins & Hernán, 2008). Let $Y[a]$ be the potential outcome, either factual or counterfactual, for treatment $a \in \mathcal{A}$. Using the dataset $\mathcal{D}$ we learn feature map $\phi(h_t, a_t)$ such that:

$$\phi(h_t, a_t) = \mathbb{E}[Y_{t+1}[a_t] \mid h_t], \tag{1}$$

where $\mathbb{E}[Y_{t+1}[a_t] \mid h_t]$ is the potential outcome for taking action $a_t$ at time $t$ given the history $h_t$. For the factual action $a_t$, assigned under policy $\pi(\cdot \mid h_t)$, the factual outcome is $x_{t+1}$ and this is the same as the potential outcome $\mathbb{E}[Y_{t+1}[a_t] \mid h_t]$. The potential outcomes for the other actions $a_t \in \mathcal{A}$ are the counterfactual ones and they allow us to understand what would happen to the patient if they receive a different treatment $a_t$. To identify the potential outcomes from the batch data we make the standard assumptions of consistency, positivity and no hidden confounders as described in Appendix B. No hidden confounders means that we observe all variables affecting the action assignment and potential outcomes. Overlap means that at each timestep, every action has a non-zero probability and can be satisfied in this setting by having a stochastic expert policy. These assumptions are standard across methods for estimating counterfactual outcome (Robins et al., 2000; Schulam & Saria, 2017; Bica et al., 2020a). Note that these assumptions are needed to be able to reliably perform causal inference using observational data. However, they do not constrain the batch IRL set-up.

Estimating the potential outcomes from batch data poses additional challenges that need to be considered. The fact that the expert follows policies that consider the history of patient observations when deciding new actions, gives rise to time-dependent confounding bias. Standard supervised learning methods for learning $\mathbb{E}[Y_{t+1}[a_t] \mid h_t]$ from $\mathcal{D}$ will be biased by the expert policy used in the observational dataset and will not be able to correctly estimate the counterfactual outcomes under alternative policies (Schulam & Saria, 2017). Methods for adjusting for the confounding bias involve

using either inverse probability of treatment weighting (Robins et al., 2000; Lim et al., 2018) or building balancing representations (Bica et al., 2020a). Refer to Appendix B for more details.

In the sequel, we consider the model for estimating counterfactuals as a black box such that the feature map $\phi(h_t, a_t)$ represents the effect of taking action $a_t$ for history $h_t$. The reward is then:

$$R(h_t, a_t) = w \cdot \phi(h_t, a_t) = w \cdot \mathbb{E}[Y_{t+1}[a_t] \mid h_t] \tag{2}$$

Defining the reward function using counterfactuals gives an interpretable parameterization of doctor behavior: It allows us to interpret their behavior with respect to the importance weights implicitly assigned to the effects of their actions. This enables describing the relative trade-offs in treatment decisions. Note that we are *not* assuming that the experts themselves actually compute these quantities (nor that they explicitly adopt the same causal inference assumptions); rather, we are simply providing a way to understand how decision-makers are effectively behaving (i.e. in terms of counterfactuals).

## 4 BATCH INVERSE REINFORCEMENT LEARNING USING COUNTERFACTUALS

Max-margin IRL (Abbeel & Ng, 2004) starts with an initial random policy $\pi$ and iteratively performs the following three steps to recover the expert policy and its reward weighs: (1) estimate feature expectations $\mu^\pi$ of candidate policy $\pi$, (2) compute new reward weights $w$ and (3) find new candidate policy $\pi$ that is optimal for reward function $R(h_t, a_t) = w \cdot \phi(h_t, a_t)$. This approach finds a policy $\tilde{\pi}$ that satisfies $\|\mu^{\pi_e} - \mu^{\tilde{\pi}}\|_2 < \epsilon$ such that $\tilde{\pi}$ has an expected value function close the expert policy.

The expert feature expectations can be estimated empirically from the dataset $\mathcal{D}$ using:

$$\mu^{\pi_E} = \frac{1}{N} \sum_{i=1}^{N} \sum_{t=0}^{T^i} \gamma^t \phi(h_t^i, a_t^i). \tag{3}$$

In the batch setting, we cannot estimate the feature expectations of candidate policies by taking the sample mean of on-policy roll-outs: $\mu^{\tilde{\pi}} \neq \frac{1}{N} \sum_{i=1}^{N} \sum_{t=0}^{T^i} \gamma^t \phi(h_t^i, \pi(h_t^i))$. To address this off-policy nature of estimating feature expectations, we introduce a new method that leverages the estimated counterfactuals. We also make use of the counterfactuals to learn optimal policies for different reward weights. Figure 2 illustrates how we integrate "what if" reasoning into batch IRL.

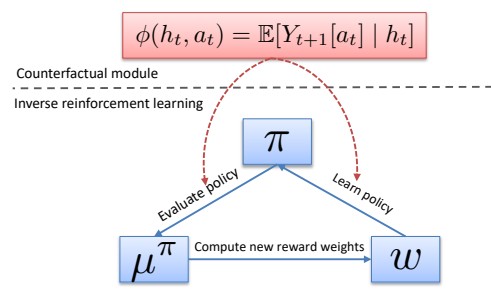

Figure 2: Counterfactual inverse reinforcement learning (CIRL). Counterfactuals are used to define $\phi(h, a)$, to estimate feature expectations $\mu^\pi$ of candidate policy $\pi$ in batch setting and to learn optimal policy for reward weights $w$.

### 4.1 COUNTERFACTUAL $\mu-$LEARNING

Similar to the approach proposed by Klein et al. (2012); Lee et al. (2019), we consider a history-action feature expectation defined as follows $\mu^\pi(h, a) = \mathbb{E}[\sum_{t=0}^{\infty} \gamma^t \phi(H_t, A_t) | \pi, H_0 = h, A_0 = a]$, where the first action $a$ can be chosen randomly and for $t \geq 1$, $A_t \sim \pi(\cdot \mid H_t)$. This can be re-written as:

$$\mu^\pi(h, a) = \phi(h, a) + \mathbb{E}_{h', a' \sim \pi(\cdot|h')}[\sum_{t=1}^{\infty} \gamma^t \phi(H_t, A_t) \mid \pi, H_1 = h', A_1 = a'] \tag{4}$$

$$= \phi(h, a) + \gamma \mathbb{E}_{h', a' \sim \pi(\cdot|h')}[\mu^\pi(h', a')], \tag{5}$$

where $h'$ is the next history. Notice the analogy between $\mu^\pi(h, a)$ and the action-value function:

$$Q^\pi(h, a) = R(h, a) + \mathbb{E}_{h', a' \sim \pi(\cdot|h')}[\sum_{t=1}^{\infty} \gamma^t R(H_t, A_t) \mid \pi, H_1 = h', A_1 = a'] \tag{6}$$

$$= R(h, a) + \gamma \mathbb{E}_{h', a' \sim \pi(\cdot|h')}[Q^\pi(h', a')], \tag{7}$$

that allows us to use temporal difference learning to estimate feature expectations (Sutton et al., 1998). Existing methods for estimating feature expectations fall into two extremes: (1) model-based (online) IRL approaches learn a model of the world and then use the model as a simulator to obtain on-policy roll-outs (Abbeel & Ng, 2004) and (2) batch IRL approaches use Q-learning (or alternative methods)

---

**Algorithm 1** (Batch, Max-Margin) CIRL

---

1: **Input**: Batch dataset $\mathcal{D}$, max iterations $n$, convergence threshold $\epsilon$,
    feature map $\phi(h_t, a_t) = \mathbb{E}[Y_{t+1}[a_t]|h_t]$
2: $\mu^{\pi_E} \leftarrow$ compute $\pi_E$'s feature expectations (Equation 3)
3: $w_0 \leftarrow$ random initial reward weights, $\pi_0 \leftarrow$ compute optimal policy for $R_0 = w_0 \cdot \phi$
4: $\mu^{\pi_0} \leftarrow$ compute $\pi_0$'s feature expectations                 (counterfactual $\mu$-learning)
5: $\Pi = \{\pi_0\}, \Delta = \{\mu^{\pi_0}\}, \bar{\mu}_0 = \mu^{\pi_0}$
6: **for** $k = 1$ to $n$ **do**
7:    $w_k = \mu^{\pi_E} - \bar{\mu}_{k-1},\ \pi_k \leftarrow$ compute optimal policy for $R_k = w_k \cdot \phi$
8:    $\mu^{\pi_k} \leftarrow$ compute $\pi_k$'s feature expectations            (counterfacual $\mu$-learning)
9:    $\Pi = \Pi \cup \{\pi_k\}, \Delta = \Delta \cup \{\mu^{\pi_k}\}$
10:   Orthogonally project $\mu^{\pi_E}$ onto line through $\bar{\mu}_{k-1}, \mu^{\pi_k}$:
$$\bar{\mu}_k = \frac{(\mu^{\pi_k} - \bar{\mu}_{k-1})^T(\mu^{\pi_E} - \bar{\mu}_{k-1})}{(\mu^{\pi_k} - \bar{\mu}_{k-1})^T(\mu^{\pi_k} - \bar{\mu}_{k-1})}(\mu^{\pi_k} - \bar{\mu}_{k-1}) + \bar{\mu}_{k-1}, \qquad t = \|\mu^{\pi_E} - \bar{\mu}_k\|_2$$
11:   **if** $t < \epsilon$ **then** break
12: **end for**
13: $K = \arg\min_{k:\mu^{\pi_k} \in \Delta} \|\mu^{\pi_E} - \mu^{\pi_k}\|_2,\ \tilde{R}(h, a) = w_K \cdot \phi(h, a)$
14: **Output**: $\tilde{R}, \Delta, \Pi$

---

for off-policy evaluation (Lee et al., 2019), and can only be used to evaluate policies similar to the expert policy and require warm start. In our case, the counterfactual model allows us to compute $h' = (h, a, \mathbb{E}[Y(a)|h])$ for any $h \in \mathcal{D}$ and any arbitrary action $a$. Thus, we propose counterfactual $\mu$-learning, a novel method for estimating feature expectations that uses these counterfactuals as part of temporal difference learning with 1-step bootstrapping. This approach falls in-between (1) and (2) and allows us to estimate feature expectations for any candidate policy $\pi$ in the batch IRL setting.

The counterfactual $\mu$-learning algorithm learns the $\mu$-values for policy $\pi$ iteratively by updating the current estimates of the $\mu$-values with the feature map plus the $\mu$-values obtained by following policy $\pi$ in the new counterfactual history $h' = (h, a, \mathbb{E}[Y[a]|h])$:

$$\hat{\mu}^\pi(h, a) \leftarrow \hat{\mu}^\pi(h, a) + \alpha(\phi(h, a) + \gamma\mathbb{E}_{a' \sim \pi(\cdot|h')}[\hat{\mu}^\pi(h', a')] - \hat{\mu}^\pi(h, a)), \tag{8}$$

where $\alpha$ is the learning rate. We use a recurrent network with parameters $\theta$ to approximate $\hat{\mu}^\pi(h, a \mid \theta)$ and we train it by minimizing the sequence of loss functions $\mathcal{L}_i$ which changes for every iteration $i$:

$$\mathcal{L}_i(\theta_i) = \mathbb{E}_{h \sim \mathcal{D}}[\||y_i - \hat{\mu}^\pi(h, a \mid \theta_i)\||_2] \qquad\qquad \theta_{i+1} \leftarrow \theta_i + \alpha\nabla(\mathcal{L}_i(\theta_i)), \tag{9}$$

where the action $a$ can be chosen randomly from $\mathcal{A}$ and $y_i = \phi(h, a) + \gamma\mathbb{E}_{a' \sim \pi(\cdot|h')}[\hat{\mu}^\pi(h', a' \mid \theta_{i-1})]$ is the target for iteration $i$. The parameters for the previous iteration $\theta_{i-1}$ are held fixed when optimizing $\mathcal{L}_i(\theta_i)$. Refer to Appendix C for full details of the counterfactual $\mu$-learning algorithm. The feature expectations for the policy $\pi$ are given by $\hat{\mu}^\pi = \mathbb{E}_{H_0, A_0 \sim \pi(\cdot|H_0)}[\hat{\mu}^\pi(H_0, A_0)]$, which can be estimated empirically from the observational dataset $\mathcal{D}$ using $\hat{\mu}^\pi = \frac{1}{N}\sum_{i=1}^{N}\sum_{a \in \mathcal{A}}\hat{\mu}^\pi(h_0^i, a)\pi(a \mid h_0^i)$.

## 4.2 FINDING OPTIMAL POLICY FOR GIVEN REWARD WEIGHTS

During each iteration of max-margin IRL, we obtain a candidate policy to evaluate by finding the optimal policy for a given vector of reward weights. We use deep recurrent $Q$-learning (Hausknecht & Stone, 2015), a model-free approach for learning $Q$-values for reward function $R(h, a) = w \cdot \phi(h, a)$. The counterfactuals are used to compute $\phi(h, a)$, and to estimate the next history for the temporal difference updates. See Appendix D for details. After estimating the $Q-$values, $Q(h, a)$ for history $h$ and action $a$, a new candidate policy is obtained using: $\pi(a \mid h) = \mathbb{1}_{a = \arg\max_{a'} Q(h, a')}$.

## 4.3 COUNTERFACTUAL INVERSE REINFORCEMENT LEARNING ALGORITHM (CIRL)

Algorithm 1 describes our proposed counterfactual inverse reinforcement learning (CIRL) method for the batch setting. CIRL is based on the projection algorithm proposed by Abbeel & Ng (2004) and iteratively updates the reward weights to minimize the margin between the expert's feature expectations and the feature expectations of intermediate policies. CIRL uses our proposed counterfactual $\mu-$learning algorithm for estimating the feature expectations of intermediate policies in an off-policy manner that is suitable for the batch setting. Compared to the algorithm for batch IRL proposed by

Lee et al. (2019) that requires the initial policy $\pi_0$ to already be similar to the expert policy, CIRL works for any initial policy $\pi$ that is optimal for the randomly initialized reward weights $w_0$.

Similarly to Choi & Kim (2011), the CIRL algorithm returns the reward function $\tilde{R}(h, a)$ that results in a policy with feature expectations closest to the ones of the expert policy. We show experimentally that the reward that yields the closest feature expectations will be similar to the true underlying reward function of the expert. CIRL returns the set of policies tried $\Pi$ and their feature expectations $\Delta$, which allows us to compute a mixing policy that would yield similar performance to the expert policy (Abbeel & Ng, 2004). Let $\tilde{\mu}$ be the closest point to $\mu^{\pi_E}$ in the convex closure of $\Delta = \{\mu^{\pi_0}, \mu^{\pi_1} \dots \mu^{\pi_k}\}$, which can be computed by solving the quadratic programming problem:

$$\min \|\mu^{\pi_E} - \mu\|_2 \text{ s.t. } \mu = \sum_i \lambda_i \mu^{\pi_i}, \lambda_i \geq 0, \sum_i \lambda_i = 1. \tag{10}$$

From the termination criteria of Algorithm 1, $\mu^{\pi_E}$ is separated from the points $\mu^{\pi_i}$ by a margin of at most $\epsilon$. Thus, the solution $\tilde{\mu}$ will satisfy $\|\mu^{\pi_E} - \tilde{\mu}\|_2 \leq \epsilon$. To obtain a policy that is close to the performance of the expert policy, we mix together the policies $\Pi = \{\pi_0, \dots \pi_k\}$ returned by Algorithm 1, where the probability of selecting $\pi_i$ is $\lambda_i$.

## 5 EXPERIMENTS

We evaluate the ability of CIRL to recover the preferences of experts over the "what if" outcomes of actions. These preferences are denoted by the magnitude of the recovered reward weights. Since we do not have access to the underlying reward weights of experts in real data, we first validate the method in a simulated environment. To show the applicability of CIRL in healthcare, we also perform a case study on an ICU dataset from the MIMIC III database (Johnson et al., 2016).

**Benchmarks.** For our proposed CIRL method, we uses the Counterfactual Recurrent Network, a state-of-the art model for estimating counterfactual outcomes in a temporal setting (Bica et al., 2020a). Note that other models for this task are also applicable (Lim et al., 2018). Refer to Appendix F for details. We benchmark CIRL against MB-IRL: model-based IRL—i.e. inverse reinforcement learning with model-based policy evaluation (e.g. Yin et al. (2016); Nagabandi et al. (2018); Kaiser et al. (2019); Buesing et al. (2018)). We consider two versions of this benchmark: MB($h$)-IRL which uses the patient history and MB($x$)-IRL which only uses the current observations to define the policy. For both MB($x$)-IRL and MB($h$)-IRL we define the reward as a weighted sum of counterfactual outcomes, but these methods use instead standard supervised methods to estimate the next history $h'$ needed for the counterfactual $\mu$-learning algorithm. These benchmarks are aimed to highlight the need of handling the bias from the time-dependent confounders and using a suitable counterfactual model, but also the importance of handling the patient history. We also compare against the deep successor feature networks (DSFN) proposed by Lee et al. (2019), which currently represents the state-of-the-art Batch IRL for the MDP setting. To show that their approach for estimating feature expectations in the batch setting is suboptimal, we extend their method to also incorporate histories in the DSFN($h$) benchmark. Implementation details of the benchmarks can be found in Appendix G.

### 5.1 EXTRACTING DIFFERENT TYPES OF EXPERT BEHAVIOUR

**Simulated environment.** We propose an environment that uses a general data simulation involving $p$-order auto-regressive processes. To analyze different types of expert behaviour (e.g. treating more/less aggressively) we simulate data for patient features representing disease progression ($x$), e.g. tumour volume and side effects ($z$) and action ($a$) indicating the binary application of treatment. For time $t$, we model the evolution of patient covariates according to the treatments as follows:

$$x_t = \frac{1}{p} \sum_{i=1}^{p} x_{t-i} - 2.5 \sum_{i=1}^{p} a_{t-i} + 0.5p + \epsilon \qquad z_t = \frac{1}{p} \sum_{i=1}^{p} z_{t-i} + 0.5 \sum_{i=1}^{p} a_{t-i} - p + \eta \tag{11}$$

where $p = 5$ and $\epsilon \sim \mathcal{N}(0, 0.1^2)$, $\eta \sim \mathcal{N}(0, 0.1^2)$ are noise terms. The initial values for the features are sampled as follows: $x_0 \sim \mathcal{N}(30, 5)$ and $z_0 \sim \mathcal{N}(2, 1)$. We set $x_{max} = 50$ and $z_{max} = 15$. The trajectory of the patient terminates when either $x_t = 0$, $x_t \geq x_{max}$, $z_t \geq z_{max}$ or $t \geq 20$. The tumour volume $x_t$, denoting the disease progression, decreases when we give treatment and increases otherwise. Conversely, the side effects $z_t$, increase when we give treatment and decrease otherwise. We define a linear reward for taking action $a_t$ given history $h_t = (x_{0:t}, z_{0:t}, a_{0:t-1})$ as follows:

$$R(h_t, a_t) = w_1 \frac{x_{t+1}}{x_{max} - x_{min}} + w_2 \frac{z_{t+1}}{z_{max} - z_{min}} \tag{12}$$

where $w = [w_1, w_2]$, $||w||_1 \leq 1$ and $x_{t+1}$ and $z_{t+1}$ are simulated according to equations 11 to take into account the effect of action $a_t$ for history $h_t$. The features are normalized to $[0, 1]$. The best scenario for a patient is when both the side effects and tumour volume are zero and a doctor attempting to achieve this will have negative reward weights. However, different settings of the reward weights will result in different expert behaviours. For instance, for $w_1 = -0.7$ and $w_2 = -0.3$, the expert policy will focus more on the disease progression and will consequently treat more aggressively, while this behaviour will be reversed for reward weights set to $w_1 = -0.3$ and $w_2 = -0.7$. We used deep recurrent $Q-$learning (Hausknecht & Stone, 2015) to find a stochastic expert policy that optimizes the reward function for different settings of the weights. The batch dataset $\mathcal{D}$ consists of 10000 trajectories sampled from the expert policy. Refer to Appendix E for more details.

**Recovering decision making preferences of experts**. We first evaluate the benchmarks on their ability to recover the weights of the reward function optimized by the expert for the experimental setting with $\gamma = 0.99$, $w_1 = -0.3$ and $w_2 = -0.7$. Note that DSFN does not provide interpretable reward weights, and thus cannot be used for understanding the trade-offs in the expert behavior. We train each benchmark 10 times and we plot in Figure 3 the reward weights obtained for the different iterations. We show that our proposed CIRL method performs best at recovering the preferences of the expert, which in this case is to treat less aggressively. While the

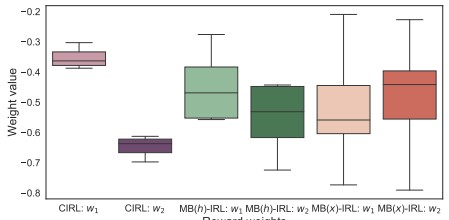

Figure 3: Reward weights recovered by benchmarks over 10 runs. The weights of the expert are $w_1 = -0.3$ and $w_2 = -0.7$.

MB($h$)-IRL method also recovers the correct trade-offs in the expert behavior, the computed weights have a much higher variance. Conversely, the MB($x$)-IRL method, which does not consider the patient history fails to recover the underlying weights of the expert policy.

**Matching the expert policy**. We evaluate the benchmarks' ability to recover policies that match the performance on the expert policy for two settings of the discount factor $\gamma \in \{0.99, 0.5\}$. A lower $\gamma$ indicates that the expert is optimizing for the immediate effect of actions, while a higher $\gamma$ means they considered the long term effect of actions. For each $\gamma$ we learn expert policies for different reward weights and we use the expert policies to generate the batch datasets. We evaluate the policies learned by the benchmarks using two metrics: cumulative reward for running the policy in the simulated environment and accuracy on matching the expert policy (computed as described in Appendix H). We report in Tables 2 and 3 the average results and their standard error over 1000 sampled trajectories from the environment. CIRL recovers a policy that has the closest cumulative reward to the expert policy and that can best match the treatments assigned by the expert.

Table 2: Mean cumulative reward and standard deviation for running learnt policy in the environment.

| Reward weights | $\gamma = 0.99$ | | | $\gamma = 0.5$ | | |
| --- | --- | --- | --- | --- | --- | --- |
| | $w_1 = -0.3$ $w_2 = -0.7$ | $w_1 = -0.7$ $w_2 = -0.3$ | $w_1 = -0.5$ $w_2 = -0.5$ | $w_1 = -0.3$ $w_2 = -0.7$ | $w_1 = -0.7$ $w_2 = -0.3$ | $w_1 = -0.5$ $w_2 = -0.5$ |
| MB($x$)-IRL | $-3.78 \pm 0.02$ | $-4.42 \pm 0.05$ | $-4.90 \pm 0.05$ | $-4.51 \pm 0.05$ | $-4.53 \pm 0.05$ | $-4.54 \pm 0.04$ |
| MB($h$)-IRL | $-3.23 \pm 0.02$ | $-4.10 \pm 0.02$ | $-4.63 \pm 0.04$ | $-4.43 \pm 0.04$ | $-3.54 \pm 0.05$ | $-4.35 \pm 0.03$ |
| DSFN | $-3.56 \pm 0.06$ | $-4.32 \pm 0.04$ | $-3.77 \pm 0.05$ | $-4.11 \pm 0.06$ | $-3.07 \pm 0.03$ | $-4.67 \pm 0.07$ |
| DSFN($h$) | $-3.31 \pm 0.07$ | $-4.33 \pm 0.07$ | $-3.60 \pm 0.07$ | $-3.95 \pm 0.07$ | $-3.05 \pm 0.05$ | $-4.61 \pm 0.05$ |
| CIRL | $\mathbf{-2.89 \pm 0.02}$ | $\mathbf{-3.92 \pm 0.03}$ | $\mathbf{-3.41 \pm 0.05}$ | $\mathbf{-2.79 \pm 0.02}$ | $\mathbf{-2.91 \pm 0.02}$ | $\mathbf{-4.27 \pm 0.03}$ |
| Expert | $-2.72 \pm 0.02$ | $-3.61 \pm 0.02$ | $-2.81 \pm 0.01$ | $-2.65 \pm 0.02$ | $-2.36 \pm 0.01$ | $-3.97 \pm 0.03$ |

Table 3: Average accuracy and standard deviation for matching the actions in the expert policy.

| Reward weights | $\gamma = 0.99$ | | | $\gamma = 0.5$ | | |
| --- | --- | --- | --- | --- | --- | --- |
| | $w_1 = -0.3$ $w_2 = -0.7$ | $w_1 = -0.7$ $w_2 = -0.3$ | $w_1 = -0.5$ $w_2 = -0.5$ | $w_1 = -0.3$ $w_2 = -0.7$ | $w_1 = -0.7$ $w_2 = -0.3$ | $w_1 = -0.5$ $w_2 = -0.5$ |
| MB($x$)-IRL | $62.5 \pm 0.41\%$ | $61.4 \pm 0.81\%$ | $54.6 \pm 0.56\%$ | $52.4 \pm 0.63\%$ | $60.1 \pm 0.39\%$ | $71.8 \pm 0.72\%$ |
| MB($h$)-IRL | $77.8 \pm 0.31\%$ | $70.2 \pm 0.45\%$ | $71.4 \pm 0.69\%$ | $66.3 \pm 0.58\%$ | $70.2 \pm 0.71\%$ | $75.6 \pm 0.52\%$ |
| DSFN | $75.4 \pm 0.32\%$ | $68.4 \pm 0.21\%$ | $73.4 \pm 0.45\%$ | $80.2 \pm 0.37\%$ | $70.8 \pm 0.24\%$ | $69.8 \pm 0.44\%$ |
| DSFN($h$) | $76.3 \pm 0.37\%$ | $67.5 \pm 0.32\%$ | $80.6 \pm 0.54\%$ | $80.4 \pm 0.56\%$ | $71.0 \pm 0.35\%$ | $70.2 \pm 0.47\%$ |
| CIRL | $\mathbf{81.8 \pm 0.42\%}$ | $\mathbf{75.5 \pm 0.51\%}$ | $\mathbf{83.7 \pm 0.76\%}$ | $\mathbf{89.5 \pm 0.37\%}$ | $\mathbf{73.2 \pm 0.43\%}$ | $\mathbf{80.4 \pm 0.42\%}$ |

## 5.2 CASE STUDY ON REAL-WORLD DATASET: MIMIC III

Suppose we want to explain the decision-making process of doctors assigning antibiotics to patients in the ICU. For this purpose, we consider a dataset with 6631 patients that have received antibiotics during their ICU stay, extracted from the Medical Information Mart for Intensive Care (MIMIC III) database (Johnson et al., 2016).

We used CIRL, with $\gamma = 0.99$, to recover the policy and the reward weights of doctors administering antibiotics to understand their preferences over the effect of antibiotics on the patient features. The relative magnitude of the reward weights for the counterfactual outcomes of the patient features considered are illustrated in Figure 4. CIRL found that reducing temperature had the highest weight in the reward function of the expert, followed by WBC. This corresponds to known medical guidelines (Marik, 2000; Palmer et al., 2008). Sepsis is a leading cause of morbidity and mortality in the ICU, and several studies have shown that early administratio of antibiotics is crucial to decrease the risk of adverse outcomes (Zahar et al., 2011). Fever and elevated WBC are among a small subset of variables that indicate a systemic inflammatory response, concerning for the development of sepsis (Neviere et al., 2017). While these findings are not specific to bacterial infection, the risk of failing to treat a potentially serious infection often outweighs the risk of inappropriate antibiotic administration, thereby driving clinicians to prescribe antibiotics in the setting of these abnormal findings. Similarly, the decision to discontinue antibiotics is complex, but it is often supported by signs of a resolution of infection which included normalization of body temperature and a downtrending of the WBC. As such, our finding that the two highest reward weights for the administration of antibiotics in the ICU is temperature and WBC is consistent with clinical practice. Moreover, note that the model is simply identifying the factors that are driving the decision-making of the clinician represented in this dataset.

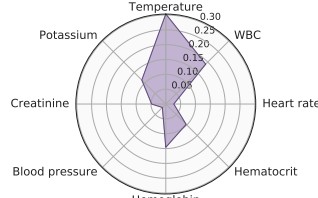

Figure 4: Radar plot of reward weights magnitude for assigning antibiotics.

|  | Accuracy |
|---|---|
| MB($x$)-IRL | $70.1 \pm 0.11\%$ |
| MB($h$)-IRL | $77.5 \pm 0.15\%$ |
| DSFN | $73.5 \pm 0.25\%$ |
| DSFN($h$) | $75.3 \pm 0.19\%$ |
| CIRL | $\mathbf{83.4 \pm 0.17}\%$ |

Table 4: Accuracy on matching expert actions.

In Table 4, to verify that explainability does not come at the cost of accuracy we also evaluate the benchmarks on matching the expert actions.

## 6 DISCUSSION

In this paper, we propose building interpretable parametrizations of sequential decision-making by explaining an expert's behaviour in terms of their preferences over "what-if" outcomes. To achieve this, we introduce CIRL, a new method that incorporates counterfactual reasoning into batch IRL: counterfactuals are used to define the feature map part of the reward function, but also to tackle the off-policy nature of estimating feature expectations in the batch setting. The reward weights recovered by CIRL indicate the relative preferences of the expert over the counterfactual outcomes of their actions. Our aim is to provide a description of behavior, i.e. how the expert is effectively behaving under our interpretable parameterization of the reward based on counterfactuals. We are not assuming that the experts actually operate under this specific (linear, in our case) model or that they compute the exact counterfactuals. Instead, our purpose is to show that we can explain an agent's behavior on the basis of counterfactuals, which is useful in that it allows us to audit them, sanity-check their policies and find variation in practice. Further discussion can be found in Appendix I.

There are several limitations of your method and directions for future work. While our method considers reward functions that are linear in the features, one way of extending it to handle more complex reward functions is to use domain knowledge to define the feature map as a non-linear function over the counterfactual outcomes. Nevertheless, these functions should be defined in a way that still allows us to obtain interpretable explanations of the expert's behaviour. Moreover, although time-invariant reward weights $w$ are standard in the IRL literature (Abbeel & Ng, 2004; Choi & Kim, 2011; Lee et al., 2019), time-variant rewards/policies have been considered in the dynamic treatment regimes (reinforcement learning) literature (Chakraborty, 2013; Zhang & Bareinboim, 2019). Thus another direction for future work would be to extend our method to consider non-stationary policies and reward weights that can change over time.

## ACKNOWLEDGMENTS

We would like to thank the reviewers for their valuable feedback. The research presented in this paper was supported by The Alan Turing Institute, under the EPSRC grant EP/N510129/1, by Alzheimer's Research UK (ARUK), by the US Office of Naval Research (ONR), and by the National Science Foundation (NSF) under grant numbers 1407712, 1462245, 1524417, 1533983, and 1722516. The authors are also grateful to Brent Ershoff for the insightful discussions and help with interpreting the medical results on MIMIC III.

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

## A    ADDITIONAL RELATED WORKS

Alternative approaches for IRL require a known MDP or POMDP model. In this context, several Bayesian approaches to IRL have also been proposed which make use of the know transition dynamics provided by the model of the environment (Ziebart et al., 2008; Ramachandran & Amir, 2007). We do not assume access to a MDP or POMDP model. Makino & Takeuchi (2012) propose a method for estimating the unknown parameters of a POMDP model of the environment; however their method is only applicable to discrete observation spaces. We do not make any assumptions about the observation space. While several model-free IRL methods have also been developed, these are also not applicable in the batch setting where we only have access to a fixed set of expert trajectories. Relative Entropy IRL (Boularias et al., 2011) requires non-expert trajectories following arbitrary policy; such experimentation to obtain non-expert trajectories is not possible in the healthcare setting. Distance Minimization IRL (DM-IRL) (Burchfiel et al., 2016) learns reward functions from scores assigned by experts to sub-optimal demonstrations. Moreover, guided cost learning proposed by Finn et al. (2016) which performs model-free maximum entropy optimization also requires the ability to sample trajectories from the environment. Similarly, adversarial IRL, proposed by Fu et al. (2017) builds upon the maximum entropy framework in Ziebart et al. (2008), but also requires the ability to execute trajectories from candidate policies.

## B    COUNTERFACTUAL ESTIMATION AND TIME-DEPENDENT CONFOUNDING

Observational patient data (e.g. electronic health records) contain information about how actions, such as treatments are performed by doctors and how they affect the patient's trajectory. Such data can be used by causal inference methods to estimate counterfactual outcomes–what would happen to the patient if the expert takes a particular action given a history of observations?

To identify the counterfactual outcomes from observational data we make the standard assumptions of consistency, positivity and no hidden confounders as described in Assumption 1 (Rosenbaum & Rubin, 1983; Robins et al., 2000). Under Assumption 1, $\mathbb{E}[Y_{t+1}[a_t] \mid h_t] = \mathbb{E}[X_{t+1} \mid a_t, h_t]$ and this can be estimated by training a regression model on the batch observational data.

**Assumption 1** (Consistency, Ignorability and Overlap). *For any individual $i$, if action $a_t$ is taken at time $t$, we observe $X_{t+1} = Y_{t+1}[a_t]$. Moreover, we have sequential strong ignorability $\{Y_{t+1}[a]\}_{a \in \mathcal{A}} \perp\!\!\!\perp a_t \mid h_t, \forall t$ and sequential overlap $\Pr(A_t = a \mid h_t) > 0, \forall t, \forall a \in \mathcal{A}$.*

The sequential strong ignorability assumption, also known as no hidden confounders, means that we observe all variables affecting the action assignment and potential outcomes. Sequential overlap means that at each timestep, every action has a non-zero probability, which can be satisfied by having a stochastic expert policy. These assumptions are standard across causal inference methods (Robins et al., 2000; Schulam & Saria, 2017; Lim et al., 2018; Bica et al., 2020a). We emphasize that these assumptions are needed to be able to reliably perform causal inference using observational data. Nevertheless, they do not constrain the batch IRL set-up.

Using observational data to estimate the counterfactual outcomes poses additional challenges in this set-up that need to be considered. In particular, direct estimation of the treatment effects is biased by the presence of time-varying confounders (Mansournia et al., 2012), which are patient covariates that influence the assignment of treatments and which, in turn, are affected by past actions.

For instance, consider that the doctor's policy of assigning treatments takes into account whether the patients' covariates denoting disease progression, such as tumor volume, has been increasing above normal thresholds for several timesteps. If these patients are also more likely to have severe side effects, without adjusting for the bias introduced by time-varying confounders, we may incorrectly conclude that the treatment is harmful to the patients (Platt et al., 2009). Using standard supervised learning methods to estimate the patients' response to treatments in this setting will be biased by the doctor's policy and will not be able to estimate the effect of treatments under different policies. This represents an issue for our batch IRL setting where we need to be able to evaluate candidate policies that are different from the expert policy. Thus, being able to correctly estimate counterfactual treatment outcomes is crucial for our method. Methods for adjusting for the confounding bias use either inverse probability of treatment weighting (IPTW) or balancing representations. The first approach, IPTW, involves creating a pseudo-population where the probability of taking action $a_t$

does not depend on $x_{0:t}$ (Robins et al., 2000; Lim et al., 2018), while the second approach builds a balancing representation of the history that is invariant to the treatment (Bica et al., 2020a). Using either approach will result in an unbiased estimation of the counterfactual outcomes. See Robins et al. (2000); Lim et al. (2018); Bica et al. (2020b) for more details about time-dependent confounding as well as for a more in-depth review of alternative methods for removing the bias from time-dependent confounders and estimating counterfactual outcomes.

## C    COUNTERFACTUAL $\mu$-LEARNING

Our counterfactual $\mu$-learning algorithm involves learning the $\mu$-values for policy $\pi$ iteratively by updating the current estimates of the $\mu$-values with the feature map plus the $\mu$-values obtained by following policy $\pi$ in the new counterfactual history $h' = (h, a, \mathbb{E}[Y[a]|h])$:

$$\hat{\mu}^\pi(h, a) \leftarrow \hat{\mu}^\pi(h, a) + \alpha(\phi(h, a) + \gamma \mathbb{E}_{a' \sim \pi(\cdot|h')}[\hat{\mu}^\pi(h', a')] - \hat{\mu}^\pi(h, a)), \tag{13}$$

where $\alpha$ is the learning rate and $\gamma$ is the discount factor.

We approximate $\hat{\mu}^\pi(h, a \mid \theta_f)$ using an RNN with parameters $\theta_f$. The RNN consists of an LSTM unit with linear output. To stabilize training we use a target network $\bar{\mu}^\pi$ with parameters $\theta_f^-$ that provide updates for the main network. Note that using a target network is standard in $Q-$learning (Mnih et al., 2013; Hausknecht & Stone, 2015). The target network $\bar{\mu}^\pi$ has the same architecture as the main network $\hat{\mu}^\pi$ and its parameters $\theta_f^-$ are updated to match $\theta_f$ every $M^-$ iterations.

The loss function for updating the $\mu$ network for estimating feature expectations at iteration $i$ is:

$$\mathcal{L}_{f,i}(\theta_i) = \mathbb{E}_{h \sim \mathcal{D}}[||y_{f,i} - \hat{\mu}^\pi(h, a \mid \theta_{f,i})||_2] \tag{14}$$

$$\theta_{f,i+1} \leftarrow \theta_{f,i} + \alpha \nabla(\mathcal{L}_i(\theta_{f,i})) \tag{15}$$

where $\alpha$ is the learning rate, $\mathcal{D}$ is the batch observational dataset and

$$y_{f,i} = \begin{cases} \phi(h, a), & \text{if patient trajectory terminates at history } h \\ \phi(h, a) + \gamma \mathbb{E}_{a' \sim \pi(\cdot|h')}[\bar{\mu}^\pi(h', a' \mid \theta_f^-)], & \text{otherwise} \end{cases} \tag{16}$$

is the target for iteration $i$ obtained using the target network $\bar{\mu}^\pi$. The action $a$ is chosen by following an $\varepsilon$-greedy policy by selecting action $a \sim \pi(\cdot \mid h)$ with probability $1 - \varepsilon$ and a random action with probability $\epsilon$. We perform $M$ training iterations. Algorithm 2 describes the full training procedure used.

Table 5 indicates the search ranges used for the various hyperparameters involved in the counterfactual $\mu$-learning algorithm. The hyperparameters were selected to ensure stability of training and convergence of the algorithm.

| Hyperparameter | Hyperparameter search range | Hyperparameter value |
|---|---|---|
| LSTM size | 64, 128, 256 | 128 |
| Batch size | 128, 256, 512 | 256 |
| Learning rate | 0.00001, 0.0001, 0.001 | 0.001 |
| Target network update $M^-$ | 100, 200, 500 | 100 |
| Min $\varepsilon$ | - | 0.0 |
| Max $\varepsilon$ | - | 0.9 |
| $\varepsilon$ decay | - | 0.00001 |
| Number of training iterations $M$ | - | 20000 |
| Optimizer | - | Adam |

Table 5: Hyperparameters for training $\mu$-network for estimating feature expectations.

---

**Algorithm 2** Counterfactual $\mu-$learning

---

1: **Input**: Batch dataset $\mathcal{D} = \{(x_0^i, a_0^i, \ldots x_{T^i-1}^i, a_{T^i-1}^i, x_{T^i}^i)\}_{i=1}^N$, policy to evaluate $\pi$, counterfactual model $\phi(h, a)$
2: Initialize feature expectations network $\hat{\mu}^\pi$ with random weights $\theta_f$
3: Initialize target feature expectations network $\bar{\mu}^\pi$ with random weights $\theta_f^- = \theta_f$
4: **for** $i = 1$ to $M$ **do**
5:     Sample random minibatch $\mathcal{B} = \{h^b\}_{b=1}^B$ of histories $h$ from $\mathcal{D}$
6:     **for** $b = 1$ to $B$ **do**
7:         With probability $\epsilon$ select random action $a^b$, otherwise select $a^b \sim \pi(\cdot \mid h^b)$
8:         Compute counterfactual histories $h^{b'} = (h^b, a^b, \mathbb{E}[Y[a^b] | h^b])$ using the counterfactual model: $\mathbb{E}[Y[a^b]|h^b] = \phi(h^b, a^b)$.
9:         Set targets:

$$y_{f,i}^b = \begin{cases} \phi(h^b, a^b), & \text{if trajectory terminates at history } h^b \\ \phi(h^b, a^b) + \gamma \mathbb{E}_{a' \sim \pi(\cdot|h^{b'})}[\bar{\mu}^\pi(h^{b'}, a' \mid \theta_f^-)], & \text{otherwise} \end{cases} \tag{17}$$

10:     **end for**
11:     Perform gradient descent step on $\sum_{b=1}^B ||y_{f,i}^b - \hat{\mu}^\pi(h^b, a^b \mid \theta_{f,i})||_2$ with respect to parameters $\theta_f$
12:     **if** $i \mod M^- = 0$ **then** $\theta_f^- = \theta_{f,i}$
13: **end for**
14: $\hat{\mu}^\pi = \frac{1}{N} \sum_{i=1}^N \sum_{a \in \mathcal{A}} \hat{\mu}^\pi(h_0^i, a)\pi(a \mid h_0^i)$
15: **Output**: $\hat{\mu}^\pi$

---

## D   Finding optimal policy for given reward weights

We use deep recurrent $Q$-learning (Hausknecht & Stone, 2015) to find the optimal policy for each setting of the reward weights $R(h, a) = w \cdot \phi(h, a)$.

For history $h$, let the next history for taking action $a$ be $h' = (h, a, \mathbb{E}[Y(a) \mid h])$, where $\mathbb{E}[Y(a) \mid h]$ is estimated by the counterfactual model. Using the batch dataset $\mathcal{D}$, we learn the $Q-$values iteratively by updating the current estimate of the $Q-$values towards the reward plus the maximum $Q$-value over all possible actions in the new history $h'$:

$$Q(h, a) \leftarrow Q(h, a) + \alpha(R(h, a) + \gamma \max_{a'} Q(h', a') - Q(h, a)), \tag{18}$$

where $\alpha$ is the learning rate and $\gamma$ is the discount factor.

We approximate $Q(h, a|\theta_q)$ using an RNN parameterized by $\theta_q$. The RNN consists of an LSTM unit with linear output. We use the standard practices for training $Q-$networks (Mnih et al., 2013; Hausknecht & Stone, 2015) and we employ a target network $\bar{Q}$ with parameters $\theta_q^-$ to provide the updates for the main network and to stabilize training. The target network $\bar{Q}$ is the same architecture as the main network $Q$ and its parameters $\theta_q^-$ are updated to match $\theta_q$ every $M^-$ iterations.

We use the following loss function for the $Q-$learning update at iteration $i$:

$$\mathcal{L}_{q,i}(\theta_i) = \mathbb{E}_{h \sim \mathcal{D}}[(y_{q,i} - Q(h, a|\theta_{q,i}))^2] \tag{19}$$

$$\theta_{q,i+1} \leftarrow \theta_{q,i} + \alpha \nabla(\mathcal{L}_{q,i}(\theta_{q,i})) \tag{20}$$

where $\alpha$ is the learning rate and

$$y_{q,i} = \begin{cases} R(h, a), & \text{if patient trajectory terminates at history } h \\ R(h, a) + \gamma \max_{a'} \bar{Q}(h', a'|\theta_q^-), & \text{otherwise} \end{cases} \tag{21}$$

is the stale update target obtained from the target network $\bar{Q}$. The action $a$ is chosen by following an $\varepsilon$-greedy policy by selecting action $a = \arg\max_{a'} Q(h, a'|\theta_q)$ with probability $1 - \varepsilon$ and a random action with probability $\epsilon$. We perform $M$ training iterations. Table 6 indicates the search ranges used for the various hyperparameters involved in the deep recurrent $Q-$learnign algorithm. The hyperparameters were selected to ensure stability of training and convergence of the algorithm.

| Hyperparameter | Hyperparameter search range | Hyperparameter value |
|---|---|---|
| LSTM size | 64, 128, 256 | 128 |
| Batch size | 128, 256, 512 | 256 |
| Learning rate | 0.00001, 0.0001, 0.001 | 0.001 |
| Target network update $M^-$ | 100, 200, 500 | 100 |
| Min $\varepsilon$ | - | 0.0 |
| Max $\varepsilon$ | - | 0.9 |
| $\varepsilon$ decay | - | 0.00001 |
| Number of training iterations $M$ | - | 20000 |
| Optimizer | - | Adam |

Table 6: Hyperparameters used for training $Q$-network to find optimal policy for a setting of the reward weights.

## E  DATA SIMULATION

Using the dynamics of the simulated environment described in Equation 22, we trained a deep recurrent $Q$-learning agent (Hausknecht & Stone, 2015) to find a stochastic optimal policy that optimizes the reward function for different settings of the reward weights $w_1$ and $w_2$ in Equation 23.

Let $h_t = (x_{0:t}, z_{0:t}, a_{0:t-1})$ contain the history of the simulated patient covariates representing disease progression $x$ and side effects $z$. Let $h_{t+1} = (x_{0:t+1}, z_{0:t+1}, a_{0:t})$ be the new history after taking action $a_t$ given $h_t$ and let $r_t = R(h_t, a_t)$ be the corresponding reward.

We again learn the $Q-$values iteratively by updating the current estimate of the $Q-$values for history $h_t$ towards the reward plus the maximum $Q$-value over all possible actions in the next history $h_{t+1}$.

$$Q(h_t, a_t) \leftarrow Q(h_t, a_t) + \alpha(R(h_t, a_t) + \gamma \max_{a'} Q(h_{t+1}, a') - Q(h_t, a_t)) \tag{22}$$

where $\alpha$ is the learning rate and $\gamma$ is the given discount factor.

We approximate the action-value function $Q(h_t, a_t|\theta_e)$ using an RNN parameterized by $\theta_e$. We use an LSTM (Hochreiter & Schmidhuber, 1997) unit as part of the RNN with a liner output activation for estimating the $Q$-values. We use the standard practices for training $Q-$networks (Mnih et al., 2013; Hausknecht & Stone, 2015).

Let $\mathcal{E} = \{(h_t, a_t, r_t, h_{t+1})\}_{t=0}^E$ be the experience replay memory of maximum capacity $E$ obtained by simulating patient trajectories from the environment. During training, the $Q-$learning agent selects and executes actions in the environment using an $\varepsilon-$greedy policy that follows the greedy policy $a = \arg\max_{a'} Q(h, a'|\theta_e)$ with probability $1 - \varepsilon$ and selects random action with probability $\epsilon$. The trajectories obtained from the agent's behaviour are added to the experience replay memory which is then used to train the $Q-$network. At each training iteration, we sample a batch $B$ of examples from $\mathcal{E}$.

Moreover, we use a target network $\bar{Q}$ with parameters $\theta_e^-$ to provide the updates to the main network. The target network $\bar{Q}$ is the same architecture as the main network $Q$ and its parameters $\theta_e^-$ are updated to match $\theta_e$ every $M^-$ iterations. The purpose of the target network is to stabilize training.

We use the following loss function for the $Q-$learning update at iteration $i$:

$$\mathcal{L}_{e,i}(\theta_{e,i}) = \mathbb{E}_{(h_t, a_t, r_t, h_{t+1}) \sim \mathcal{E}}[(y_{e,i} - Q(h_t, a_t|\theta_{e,i}))^2] \tag{23}$$

$$\theta_{e,i+1} \leftarrow \theta_{e,i} + \alpha \nabla(\mathcal{L}_{e,i}(\theta_{e,i})) \tag{24}$$

where $\alpha$ is the learning rate and

$$y_{e,i} = \begin{cases} r_t, & \text{if patient trajectory terminates at timestep } t \\ r_t + \gamma \max_{a'} \bar{Q}(h_{t+1}, a'|\theta_e^-), & \text{otherwise} \end{cases} \tag{25}$$

is the stale update target obtained from the target network $\bar{Q}$. We perform $M$ training iterations.

Table 7 indicates the search ranges used for the various hyperparameters involved in the deep recurrent $Q-$learnign algorithm. We selected hyperparameters based on the cummulative reward obtained from the learnt optimal greedy policy in the simulated environment.

| Hyperparameter | Hyperparameter search range | Hyperparameter value |
|---|---|---|
| LSTM size | 64, 128, 256 | 128 |
| Experience replay memory capacity E | 10000, 50000 | 10000 |
| Batch size | 128, 256, 512 | 256 |
| Learning rate | 0.00001, 0.0001, 0.001 | 0.001 |
| Target network update $M^-$ | 100, 200, 500 | 200 |
| Min $\varepsilon$ | - | 0.0 |
| Max $\varepsilon$ | - | 0.9 |
| $\varepsilon$ decay | - | 0.00005 |
| Number of training iterations $M$ | - | 40000 |
| Optimizer | - | Adam |

Table 7: Hyperparameters used for training $Q$-network to solve the simulated environment.

After training the $Q-$network we create a batch dataset $\mathcal{D}$ containing 10000 trajectories from the following stochastic expert policy:

$$\pi(a_t \mid h_t) = \text{Bernoulli}(\text{sigmoid}(\kappa Q(h_t, a_t)). \tag{26}$$

where $\kappa$ is a parameter that introduces time-dependent confounding bias in the expert policy.

## F  IMPLEMENTATION DETAILS CIRL

The CIRL algorithm replies on using a counterfactual model to estimate the potential outcomes $\mathbb{E}[Y[a] \mid h]$ which are used to define the reward and to estimate the feature expectations. For this, we use the Counterfactual Recurrent Network (Bica et al., 2020a) which removes the bias from the time-dependend counfounders by building a balancing representation that is invariant to the treatment received by the patient at each timestep. Refer to (Bica et al., 2020a) for details about the model architecture. Note that alternative counterfactual models can be used for this purpose (Lim et al., 2018).

For each simulated batch observational dataset, we use 9000 samples for training the Counterfactual Recurrent Network and 1000 for validation (hyperparameter optimization). We perform hyperparameter optimization for the counterfactual model using the hyperparameter search ranges described in Table 8. We select the model that has the lowest error in estimating the factual outcomes on the validation dataset.

Table 8: Hyperparameter search range for Counterfactual Recurrent Network. C is the size of the input and R is the size of the balancing representation built by the Counterfactual Recurrent Network.

| Hyperparameter | Hyperparameter search range |
|---|---|
| Iterations of Hyperparameter Search | 20 |
| Learning rate | 0.01, 0.001, 0.0001 |
| Minibatch size | 64, 128, 256 |
| RNN hidden units | 0.5C, 1C, 2C, 3C, 4C |
| Balancing representation size | 0.5C, 1C, 2C, 3C, 4C |
| FC hidden units | 0.5R, 1R, 2R, 3R, 4R |
| RNN dropout probability | 0.1, 0.2, 0.3, 0.4, 0.5 |

In the CIRL algorithm, we set the maximum number of iterations to 50 and the convergence threshold $\epsilon$ to 0.001.

The experiments were run on a system with 2 NVIDIA K80 Tesla GPUs, 12CPUs, and 112GB of RAM.

## G    Implementation details benchmarks

For benchmarks, we integrate as part of the proposed batch inverse reinforcement learning algorithm model-based policy evaluation reinforcement learning methods that use standard supervised learning approaches to estimate the next history (Nagabandi et al., 2018). These methods use standard supervised learning to estimate the next history $h'$ needed for the counterfactual $\mu$-learning algorithm and for finding the optimal policy given a vector of reward weights. Our MB($h$) benchmark uses a standard RNN to estimate the next history and define the policy, while MB($x$) uses a multi-layer perceptron (MLP) that only considers the current observations for this purpose. The aim of these benchmarks is to highlight the importance of handling the the bias from the time-depdendent confounders and using a suitable counterfactual model, but also the importance of handling the patient history.

For each simulated batch observational datasets, we use 9000 of the samples for trainign and 1000 for validation. We chose hyperparameters according to the error of the models in estimating the next patient history on the validation set.

The MB($h$) benchmark receives as input the patient history $h_t$ and current action $a_t$ and estimates the next observations $x_{t+1}$ which are used to form the next history $h_{t+1}$. For this purpose, the MB($h$) benchmark uses an LSTM unit, with a fully connected layer (FC) on top with ELU activation. The hyperparameter search ranges used for this model are described in Table 9.

Table 9: Hyperparameter search range for the RNN. C is the size of the input.

| Hyperparameter | Hyperparameter search range |
|---|---|
| Iterations of Hyperparameter Search | 20 |
| Learning rate | 0.01, 0.001, 0.0001 |
| Minibatch size | 64, 128, 256 |
| RNN hidden units | 1C, 2C, 3C, 4C, 5C |
| FC hidden units | 1C, 2C, 3C, 4C, 5C |

Conversely, the MB($x$) benchmark receives as input the current patient observation $x_t$ and current action $a_t$ and is trained to estimate the next observation $x_{t+1}$. The MB($x$) uses a multi-layer perceptron with two fully connected layers and ELU activation. The hyperparameter search ranges used for this model are described in Table 10.

Table 10: Hyperparameter search range for MLP. C is the size of the input.

| Hyperparameter | Hyperparameter search range |
|---|---|
| Iterations of Hyperparameter Search | 20 |
| Learning rate | 0.01, 0.001, 0.0001 |
| Minibatch size | 64, 128, 256 |
| FC hidden units | 2C, 3C, 4C, 5C |

## H    Evaluating accuracy on matching expert actions

The accuracy on matching the expert policy is computed as follows. Consider running the expert policy $\pi_e$ in the environment and obtaining a test dataset with N trajectories: $D_t = \{\zeta^i\}_{i=1}^N$, where each $\zeta^i = (x_0^i, a_0^i, \dots x_{T^i-1}^i, a_{T^i-1}^i, x_{T^i}^i)$ has length $T^i$. Let $\hat{a}_t$ be the action selected by the policy $\hat{\pi}$ recovered by CIRL for history $h_t = (x_{0:t}, a_{0:t-1})$. The accuracy for matching the expert policy is equal to $\frac{1}{N}\sum_{i=1}^N(\frac{1}{T^i}(\sum_{t=1}^{T^i}\mathbb{I}[a_t == \hat{a}_t]))$.

## I LIMITATIONS AND INTERACTION BETWEEN COUNTERFACTUAL MODULE AND CIRL

We illustrate in Figure 5 how the different components in our method interact and which are their limitations.

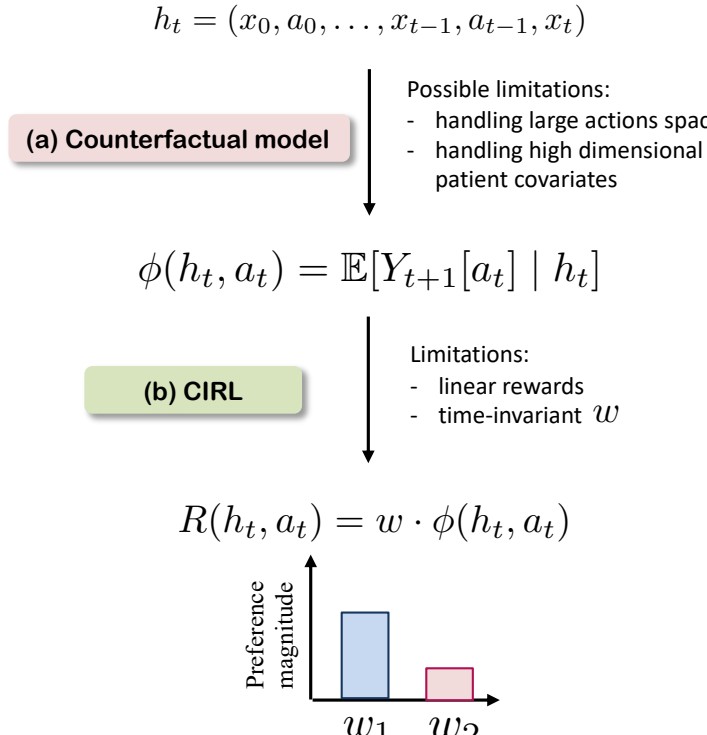

Figure 5: The two different components part of our method for obtaining interpretable parametrizations of decision making: (a) Counterfactual model which estimates the potential outcomes for taking action $a_t$ given patient history $h_t$. Note that we do not propose a new counterfactual estimation algorithm, but instead use already existing methods which could have limitations in terms of how they handle large action spaces and large patient covariates. (b) CIRL algorithm which represents the novelty of this work. We propose incorporating the estimated counterfactuals as part of a method for batch IRL that can recover the preferences of the expert over the "what if" patient outcomes. As part of the limitations of CIRL are the use of linear rewards and time-invariant reward weights $w$.

