# OpenReview forum: "Learning "What-if" Explanations for Sequential Decision-Making"
_ICLR.cc/2021/Conference — ICLR 2021 Poster_

### Official Review · AnonReviewer2 · 2020-10-27
**An interesting approach to learn from decision making**

**Rating:** 7
**Confidence:** 4

**Review:**

The paper describes how to learn cost benefit tradeoffs associated with the expert’s actions. The proposed approach integrates counterfactual reasoning into batch inverse reinforcement learning and offers a sensble framework for defining reward fuctions and tentatively explain how domain expert think and act. The framework is developed for those cases where active interaction with the system under study, i.e., experimentation, is not possible, which is very often the case in healtcare.
The paper estimates the effects of different decisions by exploiting the concept of counterfactual to accommodate settings where the policy applied by the expert depends on histories of observations rather than just current states of the system under consideration.
----------------------------------------------------------------------------------------------------------------------
Reason for Score:
Overall, I vote for accepting. I like the methodological framework, it is well structured and convincing, even if the basic assumprtions required to make the proposed approach working are strict. Indeed, in many situations it is not clear whether these assumptions are satisfied or not. Furthermore, assumptions can not be tested. The problem tackled is extremely challenging from the theorectical point of view and to the best of my knowledge this is the first paper which tries to explain sequential decisions through counterfactual reasoning and to tackle the batch IRL problem in partially-observable environments.
----------------------------------------------------------------------------------------------------------------------
Pros.
1) tackles a very relevant problem, both theoretically and practically.
2) well structured and written.
3) methods sound and convincing.
3) numerial experiments are well designed and results seem to confirm the effectiveness of the proposed appproach.
----------------------------------------------------------------------------------------------------------------------
Cons.
1) at page 3 when introducing the value function of a policy, the choice of V is not that clever because early in the paper you let V to be the volume of tumor.
2) please provide quantitative description of how accuracy is computed in numerical experiments
3) the explanation components of the proposed approach should be further developed, I see potential there but at the current stage it limits to the weights and not for example tries to investigate whether these weights change during time
4) I would like to see how the proposed approach scales with the number of covariates of patients.
----------------------------------------------------------------------------------------------------------------------
Questions during rebuttal period:
Please address and clarify the cons above
----------------------------------------------------------------------------------------------------------------------
I found no typos.
----------------------------------------------------------------------------------------------------------------------

---

> ### Author Response · Authors · 2020-11-20
> **Reply to Reviewer #2 Part [1/2]**
>
> Thank you very much for your feedback which has helped us improve the paper! Please find below our answers to your comments. Please let us know if further clarifications are needed. The changes made in the revised manuscript are indicated in blue.
>
> **(1) Change of variable name**
> Thank you for pointing out the need to change the variable used to define the tumour volume. We have now updated the paper in the introduction and Figure 1 to use U to denote the tumour volume.
>
> **(2) Quantitative description of how accuracy is computed**
>
> Thank you for your suggestion! The accuracy on matching the expert policy is computed as follows. Consider running the expert policy $\pi_e$ in the environment and obtaining a test dataset with N trajectories ${\zeta}^{i}$, where each $\zeta^{i} =(x_0^{i}, a_0^{i}, \dots x_{T^{i}-1}^{i}, a_{T^{i}-1}^{i}, x_{T^{i}}^{i})$ has length $T^{i}$. Let $\alpha_t$ be the action selected by the policy $\hat{\pi}$ recovered by CIRL for history $h_t = (x_{0:t}, a_{0:t-1})$. The accuracy for matching the expert policy is equal to $\frac{1}{N} \sum_{i=1}^{N} (\frac{1}{T^{i}} (\sum_{t=1}^{T^{i}} \mathbb{I}[a_t == \alpha_t]))$. We have now included a quantitative description for how the accuracy is computed in Appendix H.
>
>
> **(3) Explanation components**
>
> We agree that the notion of “explainability” would benefit from clarification. Importantly, the explainability of our method is *not* simply due to the use of reward weights (which is common to most IRL methods). Instead, in adapting the IRL formalism, we are driven by two considerations:
>
> - First, our central thesis is that humans make decisions by first considering the different outcomes that would result from different actions. Hence in the reward function, our features are precisely these *counterfactual outcomes*, so that we can explain how they weigh the *tradeoffs* between these outcomes.
>
> - Second, the *functional form* has to be simple enough to be an intuitive explanation. While we can certainly incorporate nonlinearities/black-box functions (see the plethora of IRL formalisms in the literature), doing so would be counter to our purpose (e.g. a piecewise linear function of counterfactuals is already far less interpretable).
>
> This advantage is best explained by comparison with regular IRL, which also recovers generic weights.
>
> - Consider what a regular IRL method does (e.g. in Lee et al. [1]): It learns a **single** mapping $(s_t, a_t) |--> R(s_t, a_t)$ directly, often as a (non-interpretable) black-box function.
>
> - Instead, here we decompose learning into **two steps**: The counterfactual estimator learns the mapping $(s_t, a_t) |--> s_{t+1}$, and then with counterfactual $\mu$-learning the CIRL procedure learns the mapping $s_{t+1} |--> R(s_{t+1})$. The advantage is clearly not just in terms of *learning* (i.e. the IRL procedure is no longer tasked with learning a single, complex function), but also *interpretability* in a medically-relevant sense (i.e. the weights directly correspond to the *tradeoffs* between clearly-defined counterfactual variables).
>
> Finally, while we recognize that the time-invariance of w is a limitation of current approaches, it is standard in the literature [1-3]. Although time-variant rewards/policies have been considered for dynamic treatment regimes (reinforcement learning) [4, 5], to the best of our knowledge, there are currently no methods that address this case for the inverse RL/preference learning setting considered in this paper. That said, we do believe that future work should definitely explore this inverse problem in the setting of non-stationary policies. Therefore we have added a discussion about this aspect in the paper. We thank the reviewer for the suggestion!
>
> We have further improved our discussion section to better highlight the explanation components of our method and also to highlight the limitations of our approach.

---

> > ### Author Response · Authors · 2020-11-20
> > **Reply to Reviewer #2 Part [2/2]**
> >
> > **(4) Number of covariates**
> >
> > In brief: CIRL scales just as well as any similar IRL method. Allow us to explain:
> >
> > - In terms of incorporating *histories of covariates* (i.e. used by the counterfactual model as input): As a deep recurrent network, the counterfactual recurrent network is naturally able to handle large numbers of patient covariates.
> >
> > - In terms of computing *feature expectations* (i.e. by the counterfactual $\mu$-learning algorithm): Since counterfactual $\mu$-learning is designed in direct analogy to standard Q-learning (see equations (4)-(7)), it is therefore not any more/less challenging to scale than (1) existing methods for estimating feature expectations or (2) methods for Q learning with function approximation in general.
> >
> > Please refer to the new Appendix I in the revised paper which highlights how the different components presented in the paper interact.
> >
> > Most importantly, however, note that our goal is to enable *interpretation* of decision-making behavior. To this end, in practice we are often interested in asking what the tradeoffs are between a small number of important covariates of interest. In this sense, in practice we do not envision CIRL being used to blindly recover an extremely high-dimensional weight vector (which would be counter to the aim of an intuitive explanation).
> >
> > **References**
> >
> > [1] Donghun Lee, Srivatsan Srinivasan, and Finale Doshi-Velez.  Truly batch apprenticeship learning with deep successor features. InProceedings of the Twenty-Eighth International Joint Conference on Artificial Intelligence, IJCAI-19
> >
> > [2] Pieter Abbeel and Andrew Y Ng.  Apprenticeship learning via inverse reinforcement learning.  InProceedings of the twenty-first international conference on Machine learning, pp.  1. ACM, 2004.
> >
> > [3] Jaedeug Choi and Kee-Eung Kim. Inverse reinforcement learning in partially observable environments.Journal of Machine Learning Research, 12(Mar):691–730, 2011
> >
> > [4] Chakraborty, Bibhas. Statistical methods for dynamic treatment regimes. Springer, 2013.
> >
> > [5] Zhang, Junzhe, and Elias Bareinboim. "Near-optimal reinforcement learning in dynamic treatment regimes." Advances in Neural Information Processing Systems. 2019.

---

> ### Author Response · Authors · 2020-11-23
> **Follow-up**
>
> Dear reviewer,
>
> We would like to thank you once again for your useful comments and constructive feedback on our paper. Please let us know if our revised manuscript and replies have addressed your concerns. If you have any additional comments, we are very eager to address them.
>
> Thank you very much!

---

### Official Review · AnonReviewer1 · 2020-10-29
**Learning explanable "reward" from the expert demonstration while considering the counterfactual questions**

**Rating:** 7
**Confidence:** 3

**Review:**

Summary:
This paper proposed a way to learn explanations of expert decisions by modeling their reward function. Different from standard IRL, this main focus is on modeling the preference in the logged decision to answer the "what-if" (aka counterfactual reasoning) question. The proposed method considered the expert can take decisions based the history instead of one observation. The proposed approach is run on a simulated and a real medical dataset, showing the effectiveness of the approach.

Detailed comments:
Pros:
1. This paper is studied an important question that is related to many key applications. Learning explainable reward signals is very useful in practice, for downstream tasks such as learning an explainable AI system and for better incorporating the system with the human expert. Partial observability or the non-Markov decision policy in the data-set is very common in real-world applications but the mainstream of work in RL/IRL focuses on the MDP settings.
2. Counterfactuals are part of the nature of RL/IRL from a fixed dataset. This paper leverage the idea from counterfactual reasoning and tried to ask and answer the counterfactual in IRL. I think it is a good attempt in an important direction and also help to bring the
the gap between causal inference and the RL community.

Cons:
1. Eq 4 gives an updating rule of counterfactual $\mu$: $\hat{\mu}$. As $\mu$ and $Q$ function can be defined as an expected feature/rewards. Can the definition of $\hat{\mu}$ first be written in the expected form and then in a recursive form which gives the (DP) updating rule? In general, it's better to give a more clear definition and explanation of $\hat{\mu}$ here.
2. The empirical study in the MIMIC III dataset is interesting but it seems unclear what the conclusion can be drawn from there. Except for the accuracy of matching expert actions, is there any other further explanation of the learned reward by CIRL, and how does that compare with other IRL methods?

Minor comments:
It might worth mentioning that [1] also tried to learn the model to explain the decision in (expert) data and considered partial observability. Maybe this work is more IRL-like and [1] is more like model-based RL.

[1] POPCORN: Partially Observed Prediction Constrained Reinforcement Learning, AISTATS 2020

---

> ### Author Response · Authors · 2020-11-20
> **Reply to Reviewer #1**
>
> Thank you very much for your feedback which has helped us improve the paper! Please find below our answers to your comments. Please let us know if further clarifications are needed.  The changes made in the revised manuscript are indicated in blue.
>
> **(1) Adding definition of $\mu$ in recursive form**
>
> Thank you for your suggestions which have helped us improve the clarity of the paper. We have now updated the manuscript to also define both $\mu$ and $Q$ in recursive form and thus better highlight the temporal difference update and justify equation (8) in the revised manuscript. Please refer to the newly added equations (5) and (7).
>
> **(2) MIMIC III experiments**
>
> We would like to clarify our experimental setting. (1) First, the simulated experiments are aimed to validate our CIRL method and its ability to recover the “ground-truth” reward weights in a controlled setting. (2) One the other hand, with the empirical study on MIMIC III, our aim was to highlight the applicability of the method on a more complex real dataset---for which the reward function is unknown, and for which the dependence of the expert on the patient history is also unknown and more complex. Importantly, for the MIMIC-III experiments, we certainly do not know that the underlying reward is linear; what our method does is therefore a “projection” onto the space of reward functions that are linear in the counterfactual outcomes. What we are demonstrating is that (a) this projection yields interpretable (i.e. clinically meaningful) results, and that (b) the recovered weights are consistent with existing medical knowledge.
>
> We have expanded the explanations in Section 5.2 in the revised paper to provide further clarifications.
> (Finally, please note that for real datasets such as MIMIC III it is not possible to deploy the learned policies by the IRL methods and evaluate them in terms of the cumulative reward in the environment---which is why we have only evaluated action matching).
>
> **Additional reference**
>
> Thank you very much for providing us with the additional reference! While [1] indeed proposes a method for model-based RL and does not handle the IRL setting, we have referenced to support the need for handling the batch setting and history dependent-policies in applications such as healthcare. Please refer to the added text in the Related works (Section 2).
>
> [1] POPCORN: Partially Observed Prediction Constrained Reinforcement Learning, AISTATS 2020

---

> ### Author Response · Authors · 2020-11-23
> **Follow-up**
>
> Dear reviewer,
>
> We would like to thank you once again for your useful comments and constructive feedback on our paper. Please let us know if our revised manuscript and replies have addressed your concerns. If you have any additional comments, we are very eager to address them.
>
> Thank you very much!

---

### Official Review · AnonReviewer4 · 2020-10-29
**Review for Learning What If Explanations for Sequential Decision Making**

**Rating:** 6
**Confidence:** 2

**Review:**

## Summary

The authors address the problem of comparing the effects of different treatments in medical decision making.  They formulate this as learning a reward function via inverse reinforcement learning in a batch setting, where the only admissible data is in the form of medical records.  Their goal is to develop a framework where they estimate the reward function as well as an optimal policy, and subsequently to compare alternatives (different treatments or abstaining from treatment) for a given history.  They propose to integrate counterfactual reasoning into the process of learning optimal policies and reward functions, which they argue is critical for off-policy evaluation in this setting, the results of which is their method dubbed CIRL.  They present results of two simulation studies, as well as an experiment on the MIMIC III data set.

## Strong points

1. The authors frame the value of the ability to interrogate preferences of different policies in terms of the value it would bring to comparing data-driven clinical guidelines of when & how to treat. (cf. page 1, Introduction : *Moreover, modeling the reward function of different clinical practitioners can be revealing as to their tendencies to treat various diseases more/less aggressively (Rysavy et al., 2015), which —in combination with patient outcomes—has the potential to inform and update clinical guidelines.*

2. By re-framing batch IRL as counterfactual reasoning, they cast the feature embedding component of the reward function to be learned as a function of possible actions, which yields interpretable results (cf. page 2, Contributions).


## Weak points

1. Reading over the paper, it is not clear they make their case for interpretable decision making.  The one strong piece of evidence they rely upon is a simulation study for recovering the parameters of the data generating process in a tumour volume vs harmful side-effects (cf. Figure 3), which shows that CIRL recovers the generating weights of the reward function with lower variance than comparable methods, but this may be due to the agreement of the assumptions CIRL makes on the reward function with that of the function used in their simulation.  It would be more convincing had they also tried more complex reward functions.

2. I grant that real data (cf. the MIMIC III case study of section 5.2) does not provide reward weights for comparison, but would like to see more in-depth evaluation than the scenario provided.  In this example, the highest proportional weight is given to Temperature and WBC (White Blood-Cell count, a marker of lymphocytic activity), which is consistent with medical guidelines.  But analgesics are also indicated to reduce temperature, and WBC may be high due to viral rather than bacterial infection.  To be more convincing, I would want to see more experiments for other interventions, to see how frequently CIRL is able to recover the desired outcome(s).


## Recommendation

IRL is a bit beyond my expertise, so I'll speak mostly to the advances in interpretability.

The premise in the introduction, that different reward functions of different clinical practitioners (or even different institutions) is not tested, and so it is difficult to assay how useful this would be in a real setting.  To the extent that CIRL can recover accurate estimates of the reward function weights, these are interpretable in the same manner that weights in a GLM are interpretable.  This seems sufficient for comparing the relative frequency of actions chosen by experts given the same history, but I am not convinced it provides an explanation for why experts make their decision (cf. arxiv.org/abs/1606.03490)

The ability of the method's claim as explainable AI notwithstanding, the blending of counterfactual reasoning with batch IRL is promising, and I recommend the paper be accepted.


## Questions for the authors

- The problem formulation in section 3 is a bit light on details with respect to how counterfactual reasoning is present in the definition of the feature representation (cf. Equation 1).

- I felt as if there was some switches in nomenclature that might help people from adjacent fields better read and understand the paper.  For example, the $\mu^{\pi}(h,a)$ are described as "history action feature expectation".  However, a friend of mine who read the paper with me recognized the definition of successor representations.  Are these two names for the same quantity?

- Section 4.1 equation (5) describes how the parameters of the recurrent network are used to approximate the feature expectations, but I wonder how efficient this is as a size of the action space A?  The experiments run in the paper feature very small (binary?) actions.

- In the experiments described in Section 5 the weights in the experiments are constrained to have 1-norm < 1, but when the space of possible actions given a history is large, I wonder how can you characterize if a difference in weights between two potential actions is meaningful (i.e statistically significant)?

## Suggestions for improving the paper

- The definition for feature expectations at the end of section 4.1 are sufficiently similar to the on-policy roll-out estimation of $\mu$ hat that it bears taking some care here to explain the difference.

- The introduction to Section 4 of the paper lays out the core of Algorithm 1 very clearly, but it would help readability of the paper if the subsection titles for section 4 followed the description of the numbered list in the introduction (e.g 4.1 is entitled "Counterfactual mu-learning" rather than "Estimating feature expectations" as in the list)

---

> ### Author Response · Authors · 2020-11-20
> **Reply to Reviewer #4 Part [1/2]**
>
> Thank you very much for your feedback which has helped us improve the paper! Please find below our answers to your comments. Please let us know if further clarifications are needed. The changes made in the revised manuscript are indicated in blue.
>
> **Replies to major comments**
>
> **(1) Interpretable decision making**
>
> Please note that for the results in Figure 3, all other benchmarks use a linear reward function and define the reward as a weighted sum of counterfactual outcomes. The main difference between CIRL the compared benchmarks is that MB($h$)-IRL does not handle the time-dependent confounding bias present in observational datasets in order to estimate the counterfactuals, while MB($s$)-IRL does not take into account the patient history.
>
> In our paper, we propose to obtain interpretable explanations of the behaviour of decision-makers by recovering the trade-offs in the expert’s behaviour, i.e. their reward weights, with respect to “what if” outcomes--given the current history of observations, what would happen if an action is taken. While we acknowledge that in real-world scenarios reward functions could be arbitrarily complex, to be able to interpret the decision-making process of experts, we are essentially performing a “projection” into the space of linear reward functions defined as weighted sums of counterfactual outcomes.
>
> Moreover, please note that the experiments on the simulated environment are needed to validate our method.  For the results in Figure 3, we know the ground truth reward weights and we evaluate the ability of CIRL and of the benchmarks to correctly recover them. However, we also show its applicability to the real-world dataset extracted from MIMIC-III for which we do not have information about the form of the reward function.
>
> **(2) MIMIC III experiments**
>
> We thank the reviewer for the insightful comment concerning other etiologies of fever and leukocytosis in the ICU setting. We would like to mention that we have already been collaborating with several clinicians in the design of the paper and the experiments. Please note that fever and leukocytosis are both common in the ICU and can be due to multiple factors including infectious, inflammatory, neoplastic, and drug-induced causes. Further complicating the clinical picture is how many of the interventions in the ICU can mask a fever such as the use of dialysis and other extracorporeal circuits, and the administration of antipyretics.
>
> We chose not to evaluate the use of analgesics in this population for two reasons. First, data on the administration of analgesic medications were not available in this datatest. Second, while certain analgesics have antipyretic properties such as acetaminophen and cox inhibitors, administration of these drugs for the purpose of lowering a fever is generally not done, except for in the setting of severe hyperthermia (i.e. >40 degrees Celsius). Pharmacologically lowering a patient’s temperature in the ICU is not standard of care, and therefore we would not want to have modeled this.
>
> The reviewer is correct in that viral infections can alter white blood cell counts. However, as mentioned above, there are a multitude of conditions associated with leukocytosis including non-infectious pro-inflammatory states (e.g. post-surgical, pancreatitis) as well as the administration of certain drugs such as corticosteroids. In summary, fever and leukocytosis are both non-specific signs in clinical medicine.
>
> The decision as to whether to treat a patient with antibiotics is complex and involves the synthesis of clinical, laboratory, imaging, and microbiological data. Sepsis secondary to bacterial infection is a significant cause of morbidity and mortality in the ICU, and studies have found that early administration of antibiotics is paramount to successful treatment [1].  Leukocytosis and fever are early indicators of the presence of sepsis, and in the appropriate clinical context, physicians will often empirically start treatment with antibiotics with the notion that the benefit of treating a potentially serious infection outweighs the risk of antibiotic administration. The decision to stop administration of an antibiotic is similarly complex, and often involves evidence of resolution of the infection which includes reduction of fever and leukocytosis. As such, it is clinically reasonable that the rewards for the decision to administer antibiotics in this population are strongly related to WBC and temperature.
>
> In the revised manuscript, we added a discussion addressing these points so that the reader could better understand why the uncovering of these reward weights is consistent with clinical practice. We also add context to the decision making behind antibiotic administration and address your concerns regarding the other causes of fever and leukocytosis in the ICU.

---

> > ### Author Response · Authors · 2020-11-20
> > **Reply to Reviewer #4 Part [2/2]**
> >
> > **Replies to questions**
> >
> > **Counterfactual reasoning**
> >
> > In equation 1, the feature map $\phi(h_t, a_t) = \mathbb{E}[Y_{t+1}[a_t]|h_t]$ is defined in terms of the counterfactual outcome $Y_{t+1}[a_t]$ indicating what would happen to the patient if action $a_t$ is taken given history $h_t$. This enables us to define the reward function as a weights sum of these counterfactual outcomes: $R(h_t, a_t) =   w \cdot \phi(h_t, a_t) = w \cdot \mathbb{E}[Y_{t+1}[a_t] \mid h_t]$.
> >
> > **Feature expectations and successor representations**
> >
> > In the batch IRL literature, $\mu^{\pi}(h, a)$ is known as the feature expectations and represents the expected cumulative feature visits that are induced by a policy $\pi$ on the feature space determined by the patient histories. While successor representations are a similar concept, they are more commonly used in reinforcement learning.
> >
> > Without loss of generality, the successor representations for discrete states are defined as follows (using the definition from Vértes & Sahani [2]): $M(s_i, s_j) =   \mathbb{E}[ \sum_{t=0}^{\infty}  \gamma^{t} \mathbb{I}[s_{t} = s_j] \mid s_0 = s_i]$. We will now derive how successor representations are in fact a specific instance of feature expectations that do not depend on actions. For discrete states, feature expectations are defined as $\mu(s, a) = E[ \sum_{t=0}^{\infty} \gamma^{t} \phi(s_t, a_t) | s_0=s, a_0=a ]$. In our case, for discrete state spaces, $\phi(s_t, a_t) = \mathbb{E}[s_{t+1}[a_t] \mid s_t]$, while the successor feature is a vector where the $j$-th component is defined as $\phi_j(s_t,a_t) = \mathbb{I}[{ s_t = s_j}]$.
> >
> > In continuous states, as can be seen in equation (4) in Vértes & Sahani [2], successor representations can be defined in a similar way as feature expectations, except that here we additionally consider actions.
> >
> > **Action size**
> >
> > Please note that our counterfactual $\mu$-learning algorithm, and subsequently CIRL, do not put any restrictions on the action size that can be used. There are indeed known (and well-studied) challenges to having large action spaces with function approximation and offline learning (Sutton and Barto [3]). However, our counterfactual $\mu$-learning algorithm is designed in direct analogy to standard Q-learning (see equations (4)-(7)), and is therefore not any more/less challenging than (1) existing methods for estimating feature expectations or (2) methods for Q learning with function approximation in general.
> >
> > **Characterizing the difference in weights**
> >
> > Please note that the L1-norm constraint is simply an arbitrary choice of scaling, and does *not* limit the expressiveness of the parameterization. In fact, if $r$ is a reward function that recovers the expert’s policy, then $kr$ is also a solution for any non-negative $k$ [4]. The interpretation of the weights (hence the reward) is not affected by this arbitrarily re-scaling.
> > Hypothesis testing can be used to evaluate if the difference in weights is statistically significant. This can be achieved by re-running the algorithm several times. Consider a reward function that linearly depends on two counterfactual outcomes $R(h_t, a_t) = w_u \mathbb{E}[U_{t+1}[a_t]\mid h_t] + w_z \mathbb{E}[Z_{t+1}[a_t]\mid h_t]$. Let $\overline{w}_u$ and   $\overline{w}_z$ be the mean weights obtained after multiple re-runs of the CIRL algorithm. In this case, a Student t-test can be used to assess if these mean weights are statistically different from each other.
> >
> > **References**
> >
> > [1]  Gaieski, David F., et al. "Impact of time to antibiotics on survival in patients with severe sepsis or septic shock in whom early goal-directed therapy was initiated in the emergency department." Critical care medicine 38.4 (2010): 1045-1053.
> >
> > [2] Vértes, Eszter, and Maneesh Sahani. "A neurally plausible model learns successor representations in partially observable environments." Advances in Neural Information Processing Systems. 2019.
> >
> > [3] Sutton, Richard S., and Andrew G. Barto. Reinforcement learning: An introduction. MIT press, 2018.
> >
> > [4] Neu, Gergely, and Szepesvari, Csaba. “Apprenticeship Learning using Inverse Reinforcement Learning and Gradient Methods.” Uncertainty in Artificial Intelligence. 2007.

---

> ### Author Response · Authors · 2020-11-23
> **Follow-up**
>
> Dear reviewer,
>
> We would like to thank you once again for your useful comments and constructive feedback on our paper. Please let us know if our revised manuscript and replies have addressed your concerns. If you have any additional comments, we are very eager to address them.
>
> Thank you very much!

---

### Official Review · AnonReviewer3 · 2020-11-03
**This paper studies interpretable policy learning in batch IRL setup. Interesting problem but the paper doesn't completely live up to the expectations.**

**Rating:** 5
**Confidence:** 3

**Review:**

This paper considers the problem of Inverse Reinforcement Learning in a batch setting. In this problem, we are given several trajectories from the expert policy e.g. electronic health records, and our goal is to recover the particular reward function the expert is maximizing without any possibility to solve the forward RL problem. One of the main motivations of this paper is to provide explanations behind the expert’s decision-making.

In order to solve the batch IRL, first the paper assumes a linear reward function i.e. $R(h_t, a_t) = <w, \phi(h_t, a_t)>$. This in turn implies that the expected reward under a policy $\pi$ is the inner product between weight vector $w$ and feature expectation $\mu^{\pi}$. With this setup, the goal is to come up with a candidate policy $\pi$ which is close to the expert's policy $\pi_E$ in terms of feature expectation.

The authors propose a new algorithm to learn a policy given a dataset of trajectories from the expert policy. The main component of the algorithm is to learn the feature expectations given a candidate policy, which the authors show can be obtained through a TD learning-based algorithm. With this procedure, the main algorithm is an iterative algorithm, which chooses the current weight vector to be the residual of the observed feature expectation and the convex hull of the feature expectation of the policies found so far. The final policy is the best convex combination of the feature expectations of the candidate policies found in the algorithm.

I think the paper makes interesting contributions to the literature on IRL in the batch setting. However, I do have several questions about the experiments and some of the claims made in the paper.

1.  I don't understand what value the counterfactuals contribute to the paper. I understand that asking about outcomes for a different chosen action leads to thinking about counterfactuals, but the paper isn't using any counterfactual estimation algorithm explicitly. Furthermore, the authors should discuss when the counterfactuals $E[Y_{t+1}(a_t) | h_t]$ are well-defined. Notice that $Y_{t+1}(\cdot)$ depend on counterfactuals $Y_{t'}(\cdot)$ through history $h_t$. Are they well-defined? What if $h_t$ is continuous because some $x_{t'}$ is a continuous random variable?
2.  The authors claim that this work can handle batch IRL setting with history-dependent reward functions. But that comes with a lot of intrinsic assumptions. First of all, $w$ does not change with time, and $\phi(\cdot)$ is always $d$-dimensional. This avoids the curse of dimensionality problem as the complexity of the problem does not increase with the length of the history. This also shows up in the experiments section where the reward just depends on the history of length $5$,
3. The assumption of linear rewards and time-invariant $w$ is very critical in this paper. This should be discussed in more detail as it lets the expected reward to be written in the form of the dot product between w and feature expectation.
4.  The first simulation uses linear features and also assumes that reward just depends on the current x and v. This makes the feature expectations computations really simple. Ideally, it would have been nice to see high dimensional w, non-linear feature, and dependence of reward on the history of length > 1.
5. Finally, the authors argue that the use of counterfactuals gives an interpretable parametrization of the expert policy. I don't agree with this statement completely. Isn't it completely driven by the low-dimensional parameter $w$? The use of counterfactuals here is also different than the counterfactuals introduced by Hérnan, and Robins, 2000. Why not first pose counterfactuals for the reward functions $R(h_t,a_t)$ and then consider some restricted low-dimensional marginal models for interpretation?

In summary, I think the paper chose a great problem as finding interpretable policies in a batch IRL setting would be really useful. However, I do think the ideas are not carried out fully, and some parts of the paper can benefit significantly from better design choices.

---

> ### Author Response · Authors · 2020-11-20
> **Reply to Reviewer #3 Part [1/3]**
>
> Thank you very much for your feedback which has helped us improve the paper! Please find below our answers to your comments. Please let us know if further clarifications are needed. The changes made in the revised manuscript are indicated in blue.
>
> **(1) Counterfactuals**
>
> By way of preface, allow us to reiterate that our goal is to obtain an *interpretable parameterization* of observed decision-making behavior. To this end, we adapt the IRL formalism, with two primary considerations.
>
> - First, our central thesis is that humans make decisions by first considering the different outcomes that would result from different actions. Hence in the reward function, our features are precisely these *counterfactual outcomes*, so that we can explain how they weigh the tradeoffs between these outcomes.
>
> - Second, the *functional form* has to be simple enough to be an intuitive explanation. While we can certainly incorporate nonlinearities/black-box functions (see the plethora of IRL formalisms in the literature), doing so would be counter to our purpose (e.g. a piecewise linear function of counterfactuals is already far less interpretable).
>
> After all, the goal is to *explain*. By analogy, consider a linear regression of icecream sales on the weather. While the (“ground truth”) relationship is almost certainly not linear, this does not detract from the fact that the learned coefficients are informative as to the magnitude and direction of the relationship. Likewise, our goal is to gain insight about a doctor’s decision-making by learning the weights $w$ for counterfactuals $\phi$.
>
> (1.1) Value of Counterfactuals:
>
> To expand on the illustrative example from Figure 1 and to provide further clarifications, consider interpreting the decision making process of assigning a binary action (treatment or no treatment) given the disease progression, such as tumour volume ($U$) and side effects ($Z$) under the assignment of a binary action (treatment or no treatment). Let $\mathbb{E}[U_{t+1}[a_t]\mid h_t]$ and $\mathbb{E}[Z_{t+1}[a_t]\mid h_t]$ be the counterfactual outcomes for the two covariates if action $a_t$ is taken for history $h_t$. Note that $h_t$ contains the evolution of both covariates and past actions. We propose defining the reward as the weighted sum of the counterfactual outcomes: $R(h_t, a_t) = w_u \mathbb{E}[U_{t+1}[a_t]\mid h_t] + w_z \mathbb{E}[Z_{t+1}[a_t]\mid h_t]$. This reward function takes into account the effect of actions and allows us to directly model the preference of the expert. For instance, in this scenario, the ideal situation is when both the tumour volume and the side effects are zero so the reward weights for a doctor aiming to achieve this should be both negative. However, if $|w_u| > |w_z|$ in the underlying reward function of the doctor, it means that they are focusing more on reducing the tumour volume rather than on the associated side effects of treatments, and thus they are treating more aggressively. Alternatively, if $|w_u| < |w_z|$ it means that the side effects play a more important role and the expert is treating less aggressively. Our motivation for using counterfactuals for defining the reward comes from the idea that rational decision making considers the potential effects of actions [1].
>
> We have now incorporated this example in the paper to further clarify the value of using counterfactuals.
>
> (1.2) Counterfactual Estimation Algorithm:
>
> You are correct: Our paper is not about a new counterfactual estimation algorithm. In our framework, in fact, any counterfactual estimation algorithm can be used. Our contribution is in setting up how these counterfactuals can be integrated into the batch IRL setting to define the reward function of experts. So in our framework, we can treat the model for estimating counterfactuals as a black box such that the feature map $\phi(h_t, a_t)$ represents the effect of taking action $a_t$ for history $h_t$. Furthermore, an added benefit of using counterfactuals to define rewards in this batch setting is that it gives rise to a new method for estimating feature expectations.
>
> We have now added a new Appendix I in the revised paper to highlight how the different components presented in the paper interact.

---

> > ### Author Response · Authors · 2020-11-20
> > **Reply to Reviewer #3 Part [2/3]**
> >
> > (1.3) When are Counterfactuals Well-defined:
> >
> > The counterfactual outcomes are identifiable under the standard assumptions of consistency, overlap and no hidden confounders. Note that these assumptions are standard in the causal inference literature and do not limit our setting.
> >
> > While this was outlined in Appendix B in the original manuscript, we have now revised section 3 in the main paper to further highlight these assumptions.
> >
> > Finally, in order to compute the reward at time $t$, note that the counterfactual outcomes are computed *one step ahead*, on the basis of the *current history*, and for *all possible actions* at time $t$. The patient history contains the factual outcomes for the actions taken in the past; as factual outcomes are simply observed past outcomes, they are *always* well-defined.
> >
> > (1.4) Continuous Random Variable: Please note that our method does not treat continuous and discrete patient covariates differently and, since the models for estimating the counterfactuals and the feature expectations are based on RNNs, they can handle continuous histories. Moreover, note that in all of your experiments, the patient covariates and subsequently the history $h_t$ are continuous.
> >
> > **(2) Assumptions and experimental details**
> >
> > We would like to emphasize that, from a theoretical perspective, the assumptions made in our paper are needed to ensure the convergence of the batch IRL algorithm and are standards in the literature [2-4].
> >
> > However, note that since $\phi(\cdot)$ are the *next-step* counterfactuals, they are by definition fixed in dimensionality (i.e. the number of covariates). This is not an assumption or a limitation.
> >
> > How these counterfactuals are estimated, of course, depends on the multi-step *history*. Note that while in the simulated environment we know how many timesteps in the past are needed to define the reward, this is not the case for the experiments using the MIMIC III dataset. Our method does *not* put any restrictions on how the counterfactual outcomes and subsequently the rewards depend on the *entire history*.
> >
> > **(3) Linear rewards and time-invariant w**
> >
> > Please note that the assumption of linear rewards and time-invariant w is standard in apprenticeship learning. Nevertheless, one approach to make our method handle more flexible scenarios would be to use domain knowledge to define the feature map as a non-linear function over the counterfactual outcomes, $\phi(h_t, a_t) = f(\mathbb{E}[Y_{t+1}[a_t] \mid h_t])$, which in turn will make the reward $R(h_t, a_t) = w \cdot f(\mathbb{E}[Y_{t+1}[a_t] \mid h_t])$ non-linear in the counterfactuals. We have added discussion about this possible extension in the paper in Section 6.
> >
> > Moreover, while we recognize that time-invariant $w$ is a limitation of current approaches, it is standard in the literature [2, 3, 4]. Although time-variant rewards/policies have been considered for dynamic treatment regimes (reinforcement learning) [6, 7], to the best of our knowledge, there are currently no methods that address this case for the inverse RL/preference learning setting considered in this paper. We believe that future work should definitely explore this inverse problem in the setting of non-stationary policies and we have added a discussion about this aspect in the revised paper in Section 6. We thank the reviewer for the suggestion!
> >
> > **(4) Simulated experiment**
> > We would like to clarify the set-up of our simulated experiment. Please note that the reward function depends on the counterfactual outcomes, which in turn depend on the patient history (of up to 5 timesteps). Thus, to correctly model the reward function, the counterfactual model part of CIRL *does* to take into account the multi-step history. Moreover, in the experiments with the real data from MIMIC-III we do not have ground-truth knowledge of how much of the patient history is important and having a method that can model complex dependencies on the patient history is crucial.
> >
> > To be clear, in our paper, the policy depends on the history of observations. The reward is a function of the one-step-ahead (next) counterfactuals computed for taking each action given a history of observations. This doesn't mean that we are myopic and our policy optimizes the value function which takes into account all of the future counterfactuals. At each time t, the history $h_t$ accounts for all *past* observations and actions (i.e. forward recursion), while the policy optimizes the value function, which is the discounted sum of rewards for all time steps in the *future* (i.e. backward recursion).

---

> > > ### Author Response · Authors · 2020-11-20
> > > **Reply to Reviewer #3 Part [3/3]**
> > >
> > > **(5) Interpretable parameterization**
> > > While we agree with the reviewer that the reward weights $w$ give us the relative preferences of the experts, it is precisely the use of counterfactuals that make these preferences interpretable. For instance, in Lee et al. [4], the feature map part of the linear reward function is defined in terms of the hidden layers of a neural network. Because of this, the recovered weights for their reward function do not give an interpretable explanation of the expert’s behaviour. (See also our preface to point (1) above).
> > > Moreover, note that Robins et al., 2000 [5] propose a method for estimating the counterfactual outcomes from observational data. The focus of our paper is different: we do not introduce a new causal inference method for estimating counterfactuals, but instead propose a new batch IRL method that can use the estimated counterfactual outcomes to build inherently interpretable rewards and simultaneously addresses the cold-start problem in Lee et al. [4].
> > >
> > > **References**
> > > [1] Benjamin Djulbegovic, Shira Elqayam, and William Dale. Rational decision making in medicine: implications for overuse and underuse.Journal of evaluation in clinical practice, 24(3):655–665,2018.
> > >
> > > [2] Pieter Abbeel and Andrew Y Ng.  Apprenticeship learning via inverse reinforcement learning.  InProceedings of the twenty-first international conference on Machine learning, pp.  1. ACM, 2004.
> > >
> > > [3] Jaedeug Choi and Kee-Eung Kim. Inverse reinforcement learning in partially observable environments.Journal of Machine Learning Research, 12(Mar):691–730, 2011
> > >
> > > [4] Donghun Lee, Srivatsan Srinivasan, and Finale Doshi-Velez.  Truly batch apprenticeship learning with deep successor features. InProceedings of the Twenty-Eighth International Joint Conference on Artificial Intelligence, IJCAI-19
> > >
> > > [5] James M Robins, Miguel Angel Hernan, and Babette Brumback.  Marginal structural models and causal inference in epidemiology, 2000
> > >
> > > [6] Chakraborty, Bibhas. Statistical methods for dynamic treatment regimes. Springer, 2013.
> > >
> > > [7] Zhang, Junzhe, and Elias Bareinboim. "Near-optimal reinforcement learning in dynamic treatment regimes." Advances in Neural Information Processing Systems. 2019.

---

> ### Author Response · Authors · 2020-11-23
> **Follow-up**
>
> Dear reviewer,
>
> We would like to thank you once again for your useful comments and constructive feedback on our paper. Please let us know if our revised manuscript and replies have addressed your concerns. If you have any additional comments, we are very eager to address them.
>
> Thank you very much!

---

### Decision · Program_Chairs · 2021-01-07
**Final Decision**

**Decision:**

Accept (Poster)

**Comment:**

This paper presents a counterfactual approach to interpret aspects within a  sequential decision-making setup. The reviewers have reacted to each others' comments as well as the authors' response to their views. I am recommending acceptance of this paper, as it targets an interesting problem and presents an intriguing approach. I think the community would appreciate further discussing this paper at the conference.